# BOOLFORMER: SYMBOLIC REGRESSION OF LOGIC FUNCTIONS WITH TRANSFORMERS

## ABSTRACT

In this work, we introduce Boolformer, the first Transformer architecture trained to perform end-to-end symbolic regression of Boolean functions. First, we show that Boolformer can predict compact formulas for complex functions which were not seen during training, when provided a complete truth table. Then, we demonstrate its ability to find approximate expressions when provided incomplete and noisy observations. We evaluate the Boolformer on a broad set of real-world binary classification datasets, demonstrating its potential as an interpretable alternative to classic machine learning methods. Finally, we apply it to the widespread task of modelling the dynamics of gene regulatory networks. Using a recent benchmark, we show that Boolformer is competitive with state-of-the art genetic algorithms with a speedup of several orders of magnitude.

## 1 INTRODUCTION

Deep neural networks, in particuler those based on the Transformer architecture (Vaswani et al., 2017), have lead to breakthroughs in computer vision (Dosovitskiy et al., 2020) and language modelling (Brown et al., 2020), and have fuelled the hopes to accelerate scientific discovery (Jumper et al., 2021). However, their ability to perform simple logic tasks remains limited (Delétang et al., 2022). These tasks differ from traditional vision or language tasks in the combinatorial nature of their input space, which makes representative data sampling challenging.

Reasoning tasks have thus gained major attention in the deep learning community, either (i) with explicit reasoning in the logical domain, e.g., tasks in the realm of arithmetic (Saxton et al., 2019; Lewkowycz et al., 2022), algebra (Zhang et al., 2022) or algorithmics (Veličković et al., 2022), or (ii) implicit reasoning in other modalities, e.g., benchmarks such as Pointer Value Retrieval (Zhang et al., 2021) and Clevr (Johnson et al., 2017) for vision models, or LogiQA (Liu et al., 2020) and GSM8K (Cobbe et al., 2021) for language models. Reasoning also plays a key role in tasks which can be tackled via Boolean modelling, particularly in the fields of biology (Wang et al., 2012) and medicine (Hemedan et al., 2022).

As these endeavours remain challenging for current Transformer architectures, it is natural to examine whether they can be handled more effectively with different approaches, e.g., by better exploiting the Boolean nature of the task. In particular, when learning Boolean functions with a 'classic' approach based on minimizing the training loss on the outputs of the function, Transformers learn potentially complex interpolators as they focus on minimizing the degree profile in the Fourier spectrum, which is not the type of bias desirable for generalization on domains that are not well sampled (Abbe et al., 2022). In turn, the complexity of the learned function makes its interpretability challenging. This raises the question of how to improve generalization and interpretability of such models.

In this paper, we tackle Boolean function learning with Transformers, but we rely directly on 'symbolic regression': our **Boolformer** is tasked to directly predict a Boolean formula, i.e., a symbolic expression of the Boolean function in terms of the three fundamental logical gates (AND, OR, NOT) such as those of Figs. 1,2. As illustrated in Fig. 3, this task is framed as a sequence prediction problem: each training example is a synthetically generated function whose truth table is the input and whose formula is the target.

By moving to this setting, we decouple the symbolic task of inferring the logical formula and the numerical task of evaluating it on new inputs: the Boolformer only has to handle the first part. We show that this approach can give surprisingly strong performance both in abstract and real-world settings, and discuss how this lays the ground for future improvements and applications.

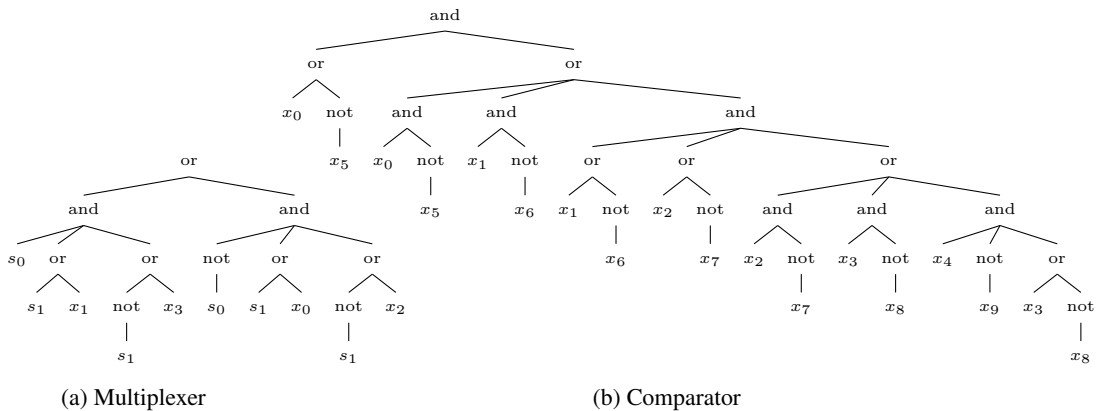

(a) Multiplexer

(b) Comparator

Figure 1: **Some logical functions for which our model predicts an optimal formula.** Left: the multiplexer, a function commonly used in electronics to select one out of four sources $x_0, x_1, x_2, x_3$ based on two selector bits $s_0, s_1$. Right: given two 5-bit numbers $a = (x_0x_1x_2x_3x_4)$ and $b = (x_5x_6x_7x_7x_9)$, returns 1 if $a > b$, 0 otherwise.

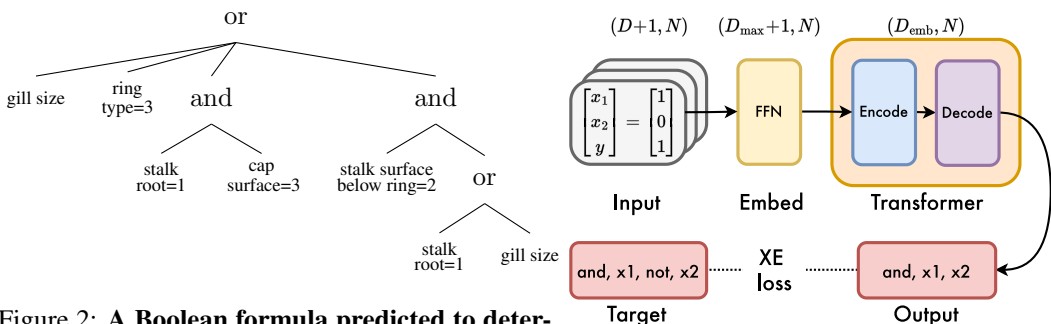

Figure 2: **A Boolean formula predicted to determine whether a mushroom is poisonous.** We considered the "mushroom" dataset from the PMLB database (Olson et al., 2017), and this formula achieves an F1 score of 0.96.

Figure 3: **Summary of our approach.** We feed $N$ points $(\boldsymbol{x}, f(\boldsymbol{x})) \in \{0, 1\}^{D+1}$ to a seq2seq Transformer, and supervise the prediction to $f$ via cross-entropy loss.

### 1.1 CONTRIBUTIONS

1. We train Transformers over synthetic datasets to perform end-to-end symbolic regression of Boolean formulas and show that given the full truth table of an unseen function, the Boolformer is able to predict a compact formula, as illustrated in Fig. 1.
2. We show that Boolformer is robust to noisy and incomplete observations, by providing incomplete truth tables with flipped bits and irrelevant variables.
3. We evaluate Boolformer on various real-world binary classification tasks from the PMLB database (Olson et al., 2017) and show that it is competitive with classic machine learning approaches such as Random Forests while providing interpretable predictions, as illustrated in Fig. 2.
4. We apply Boolformer to the well-studied task of modelling gene regulatory networks (GRNs) in biology. Using a recent benchmark, we show that our model is competitive with state-of-the-art methods with several orders of magnitude faster inference.

### 1.2 RELATED WORK

**Reasoning in deep learning**    Several papers have studied the ability of deep neural networks to solve logic tasks. Evans & Grefenstette (2018) introduce differential inductive logic as a method to learn logical rules from noisy data, and a few subsequent works attempted to craft dedicated neural architectures to improve this ability (Ciravegna et al., 2023; Shi et al., 2020b; Dong et al.,

2019). Large language models (LLMs) such as ChatGPT, however, have been shown to perform poorly at simple logical tasks such as basic arithmetic (Delétang et al., 2022; Jelassi et al., 2023), and tend to rely on approximations and shortcuts (Liu et al., 2022). Although some reasoning abilities seem to emerge with scale (Wei et al., 2022a) and can be enhanced via several procedures such as scratchpads (Nye et al., 2021) and chain-of-thought prompting (Wei et al., 2022b), achieving holistic and interpretable reasoning in LLMs remains an unsolved challenge.

**Learning Boolean functions** Leaning Boolean functions has been an active area in theoretical machine learning, mostly under the probably approximately correct (PAC) and statistical query (SQ) learning frameworks (Hellerstein & Servedio, 2007; Reyzin, 2020). More recently, Abbe et al. (2023) shows that regular neural networks learn by gradually fitting monomials of increasing degree, in such a way that the sample complexity is governed by the 'leap complexity' of the target function, i.e. the largest degree jump the Boolean function sees in its Fourier decomposition. In turn, Abbe et al. (2022) shows that this leads to a 'min-degree bias' limitation: Transformers tend to learn interpolators having least 'degree profile' in the Boolean Fourier basis, which typically lose the Boolean nature of the target and often produce complex solutions with poor out-of-distribution generalization.

**Inferring Boolean formulas** A few works have explored the paradigm of inferring Boolean formulas in symbolic form, using SAT solvers (Narodytska et al., 2018), ILP solvers (Wang & Rudin, 2015; Su et al., 2015) or LP-relaxation (Malioutov et al., 2017). However, all these works predict the formulas in conjunctive or disjunctive normal forms (CNF/DNF), which typically amounts to exponentially long formulas. In contrast, the Boolformer is biased towards predicting compact expressions[1], which is more akin to logic synthesis – the task of finding the shortest circuit to express a given function, also known as the Minimum Circuit Size Problem (MCSP). While a few heuristics (e.g. Karnaugh maps (Karnaugh, 1953)) and algorithms (e.g. ESPRESSO (Rudell & Sangiovanni-Vincentelli, 1987)) exist to tackle the MCSP, its NP-hardness (Murray & Williams, 2017) remains a barrier towards efficient circuit design. Given the high resilience of computers to errors, approximate logic synthesis techniques have been introduced (Scarabottolo et al., 2018; Venkataramani et al., 2012; 2013; Boroumand et al., 2021; Oliveira & Sangiovanni-Vincentelli, 1993; Rosenberg et al., 2023), with the aim of providing approximate expressions given incomplete data – this is similar in spirit to what we study in the noisy regime of Section 4.

**Symbolic regression** Symbolic regression (SR), i.e. the search of mathematical expressions underlying a set of numerical values, is still today a rather unexplored paradigm in the ML literature. Since this search cannot directly be framed as a differentiable problem, the dominant approach for SR is genetic programming (see (La Cava et al., 2021) for a recent review). A few recent publications applied Transformer-based approaches to SR (Biggio et al., 2021; Valipour et al., 2021; Kamienny et al., 2022; Tenachi et al., 2023), yielding comparable results but with a significant advantage: the inference time rarely exceeds a few seconds, several orders of magnitude faster than existing methods. Indeed, while the latter need to be run from scratch on each new set of observations, Transformers are trained over synthetic datasets, and inference simply consists in a forward pass.

## 2 METHODS

Our task is to infer Boolean functions of the form $f : \{0, 1\}^D \to \{0, 1\}$, by predicting a Boolean formula built from the basic logical operators: AND, OR, NOT, as illustrated in Figs. 1, 2. We train Transformers (Vaswani et al., 2017) on a large dataset of synthetic examples, following the seminal approach of Lample & Charton (2019). For each example, the input $\mathcal{D}_{\text{fit}}$ is a set of pairs $\{(\boldsymbol{x}_i, y = f(\boldsymbol{x}_i))\}_{i=1...N}$, and the target is the function $f$ as described above. Our general method is summarized in Fig. 3. Examples are generated by first sampling a random function $f$, then generating the corresponding $(\boldsymbol{x}, y)$ pairs as described in the sections below.

---

[1]Consider for example the comparator of Fig. 1: since the truth table has roughly as many positive and negative outputs, the CNF/DNF involves $\mathcal{O}(2^D)$ terms where $D$ is the number of input variables, which for $D = 10$ amounts to several thousand binary gates, versus 17 for our model.

## 2.1 GENERATING FUNCTIONS

We generate random Boolean formulas[2] in the form of random unary-binary trees with mathematical operators as internal nodes and variables as leaves. The procedure is detailed as follows:

1. **Sample the input dimension** $D$ of the function $f$ uniformly in $[1, D_{\max}]$ .
2. **Sample the number of active variables** $S$ uniformly in $[1, S_{\max}]$. $S$ determines the number of variables which affect the output of $f$: the other variables are inactive. Then, select a set of $S$ variables from the original $D$ variables uniformly at random.
3. **Sample the number of binary operators** $B$ uniformly in $[S - 1, B_{\max}]$ then sample $B$ operators from {AND, OR} independently with equal probability.
4. **Build a binary tree** with those $B$ nodes, using the sampling procedure of Lample & Charton (2019), designed to produce a diverse mix of deep and narrow versus shallow and wide trees.
5. **Negate some of the nodes** of the tree by adding NOT gates independently with probability $p_{\text{NOT}} = 1/2$.
6. **Fill in the leaves**: for each of the $B + 1$ leaves in the tree, sample independently and uniformly one of the variables from the set of active variables[3].
7. **Simplify** the tree using Boolean algebra rules, as described in App. A. This greatly reduces the number of operators, and occasionally reduces the number of active variables.

Note that the distribution of functions generated in this way spans the whole space of possible Boolean functions (which is of size $2^{2^D}$), but in a non-uniform fashion[4] with a bias towards functions which can be expressed with short formulas. To maximize diversity, we sample large formulas (up to $B_{\max} = 500$ binary gates), which are then heavily pruned in the simplification step[5]. As discussed quantitatively in App. B, the diversity of functions generated in this way is such that throughout the whole training procedure, functions of dimension $D \geq 7$ are typically encountered at most once.

To represent Boolean formulas as sequences fed to the Transformer, we enumerate the nodes of the trees in prefix order, i.e., direct Polish notation as in (Lample & Charton, 2019): operators and variables are represented as single autonomous tokens, e.g. $[\text{AND}, x_1, \text{NOT}, x_2]$[6]. The inputs are embedded using $\{0, 1\}$ tokens.

## 2.2 GENERATING INPUTS

Once the function $f$ is generated, we sample $N$ points $\boldsymbol{x}$ uniformly in the Boolean hypercube and compute the corresponding outputs $y = f(\boldsymbol{x})$. Optionally, we may flip the bits of the inputs and outputs independently with probability $\sigma_{\text{flip}}$; we consider the two following setups.

**Noiseless regime** The noiseless regime, studied in Sec. 3, is defined as follows:

- **Noiseless data:** there is no bit flipping, i.e. $\sigma_{\text{flip}} = 0$.
- **Full support:** all the input bits affect the output, i.e. $S = D$.
- **Full observability:** the model has access to the whole truth table of the Boolean function, i.e. $N = 2^D$. Due to the quadratic length complexity of Transformers, this limits us to rather small input dimensions, i.e. $D_{\max} = 10$.

**Noisy regime** In the noisy regime, studied in Sec. 4, the model must determine which variables affect the output, while also being able to cope with corruption of the inputs and outputs. During training, we vary the amount of noise for each sample so that the model can handle a variety of noise levels:

- **Noisy data:** the probability of each bit (both input and output) being flipped $\sigma_{\text{flip}}$ is sampled uniformly in $[0, 0.1]$.

---

[2]A Boolean formula is a tree where input bits can appear more than once, and differs from a Boolean circuit, which is a directed graph which can feature cycles, but where each input bit appears once at most.

[3]The first $S$ variables are sampled without replacement in order for all the active variables to appear in the tree.

[4]More involved generation procedures, e.g. involving Boolean circuits, could be envisioned as discussed in Sec. 5, but we leave this for future work.

[5]The simplification leads to a non-uniform distribution of number of operators as shown in App. A.

[6]We discard formulas which require more than 200 tokens.

- **Partial support:** the model can handle high-dimensional functions, $D_{\max} = 120$, but the number of active variables is sampled uniformly in $[0, 6]$. All the other variables are inactive.
- **Partial observability:** a subset of the hypercube is observed: the number of input points $N$ is sampled uniformly in $[30, 300]$, which is typically much smaller that $2^D$. Additionally, instead of sampling uniformly (which would cause distribution shifts if the inputs are not uniformly distributed at inference), we generate the input points via a random walk in the hypercube. Namely, we sample an initial point $x_0$ then construct the following points by flipping independently each coordinate with a probability $\gamma_{\mathrm{expl}}$ sampled uniformly in $[0.05, 0.25]$.

### 2.3 MODEL

**Embedder**  Our model is provided $N$ input points $(\boldsymbol{x}, y) \in \{0, 1\}^{D+1}$, each of which is represented by $D + 1$ tokens of dimension $D_{\mathrm{emb}}$. As $D$ and $N$ become large, this would result in very long input sequences (e.g. $10^4$ tokens for $D = 100$ and $N = 100$) which challenge the quadratic complexity of Transformers. To mitigate this, we introduce an embedder to map each input pair $(\boldsymbol{x}, y)$ to a single embedding, following Kamienny et al. (2022). The embedder pads the empty input dimensions to $D_{\max}$, enabling our model to handle variable input dimension, then concatenates all the tokens and feeds the $(D_{\max} + 1)D_{\mathrm{emb}}$-dimensional result into a 2-layer fully-connected feedforward network (FFN) with ReLU activations, which projects down to dimension $D_{\mathrm{emb}}$. The resulting $N$ embeddings of dimension $D_{\mathrm{emb}}$ are then fed to the Transformer.

**Transformer**  We use a sequence-to-sequence Transformer architecture (Vaswani et al., 2017) where both the encoder and the decoder use 8 layers, 16 attention heads and an embedding dimension of 512, for a total of around 60M parameters (2M in the embedder, 25M in the encoder and 35M in the decoder). A notable property of this task is the permutation invariance of the $N$ input points. To account for this invariance, we remove positional embeddings from the encoder. The decoder uses standard learnable positional encodings.

### 2.4 TRAINING AND EVALUATION

**Training**  We optimize a cross-entropy loss with the Adam optimizer and a batch size of 128, warming up the learning rate from $10^{-7}$ to $2 \times 10^{-4}$ over the first 10,000 steps, then decaying it using a cosine anneal for the next 300,000 steps, then restarting the annealing cycle with a damping factor of 3/2. We do not use any regularization, either in the form of weight decay or dropout. We train our models on around 30M examples; on a single NVIDIA A100 GPU with 80GB memory and 8 CPU cores, this takes about 3 days.

**Inference**  At inference time, we find that beam search is the best decoding strategy in terms of diversity and quality. In most results presented in this paper, we use a beam size of 10. One major advantage here is that we have an easy criterion to rank candidates, which is how well they fit the input data – to assess this, we use the fitting error defined below. Note that when the data is noiseless, the model will often find several candidates which perfectly fit the inputs, as shown in App. G; in this case, we select the shortest formula, i.e. that with smallest number of gates.

**Evaluation**  Given a set of input-output pairs $\mathcal{D}$ generated by a target function $f_\star$, we compute the error of a predicted function $f$ as $\epsilon_{\mathcal{D}} = \frac{1}{|\mathcal{D}|} \sum_{(\boldsymbol{x}, y) \in \mathcal{D}} 1[f(\boldsymbol{x}) = f_\star(\boldsymbol{x})]$. We can then define:

- **Fitting error:** error obtained when re-using the points used to predict the formula, $\mathcal{D} = \mathcal{D}_{\mathrm{fit}}$
- **Fitting accuracy:** defined as 1 if the fitting error is strictly equal to 0, and 0 otherwise.
- **Test error:** error obtained when sampling points uniformly at random in the hypercube outside of $\mathcal{D}_{\mathrm{fit}}$. Note that we can only assess this in the noisy regime, where the model observes a subset of the hypercube.
- **Test accuracy:** defined as 1 if the test error is strictly equal to 0, and 0 otherwise.

## 3 NOISELESS REGIME: FINDING THE SHORTEST FORMULA

We begin by the noiseless regime (see Sec. 2.2). This setting is akin to logic synthesis, where the goal is to find the shortest formula that implements a given function.

**In-domain performance** In Fig. 4, we report the performance of the model when varying the number of input bits and the number of operators of the ground truth. Metrics are averaged over $10^4$ samples from the random generator; as demonstrated in App. B, these samples have typically not been seen during training for $D \geq 7$. We observe that the model is able to recover the target function with high accuracy in all cases, even for $D \geq 7$ where memorization is impossible. We emphasize however that these results only quantify the performance of our model on the distribution of functions it was trained on, which is highly-nonuniform in the $2^{2^D}$-dimensional space of Boolean functions. We give a few examples of success and failure cases below.

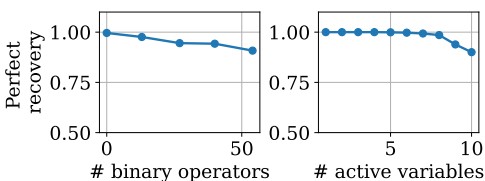

Figure 4: **Our model is able to recover the formula of unseen functions with high accuracy.** We report the fitting accuracy of our model when varying the number of binary gates and input bits. Metrics are averaged over 10k samples from the random function generator.

**Success and failure cases** In Fig. 1, we show two examples of Boolean functions where our model successfully predicts a compact formula for: the 4-to-1 multiplexer (which takes 6 input bits) and the 5-bit comparator (which takes 10 input bits). In App. D, we provide more examples: addition and multiplication, as well as majority and parity functions. By increasing the dimensionality of each problem up to the point of failure, we show that in all cases our model typically predicts exact and compact formulas as long as the function can be expressed with less than 50 binary gates (which is the largest size seen during training) and fails beyond.

Hence, the failure point depends on the intrinsic difficulty of the function: for example, Boolformer can predict an exact formula for the comparator function up to $D = 10$, but only $D = 6$ for multiplication, $D = 5$ for majority and $D = 4$ for parity as well as typical random functions (whose outputs are independently sampled from $\{0, 1\}$). Parity functions are well-known to be the most difficult functions to learn for SQ models due to their leap-complexity, and are also the hardest to learn in our framework because they require the most operators to be expressed (the XOR operator being excluded in this work).

## 4 NOISY REGIME: APPLICATIONS TO REAL-WORLD DATA

We now turn to the noisy regime, which is defined at the end of Sec. 2.2. We begin by examining in-domain performance as before, then present two real-world applications: binary classification and gene regulatory network inference.

### 4.1 RESULTS ON NOISY DATA

In Fig. 5, we show how the performance of our model depends on the various factors of difficulty of the problem. The different colors correspond to different numbers of active variables, as shown in the leftmost panel: in this setting with multiple sources of noise, we see that accuracy drops much faster with the number of active variables than in the noiseless setting.

As could be expected, performance improves as the number of input points $N$ increases, and degrades as the amount of random flipping and the number of inactive variables increase. However, our model copes relatively well with noise in general, as it displays nontrivial generalization even when we add up to 120 inactive variables and up to 10% random flipping.

### 4.2 APPLICATION TO INTERPRETABLE BINARY CLASSIFICATION

In this section, we show that our noisy model can be applied to binary classification tasks, providing an interpretable alternative to classic machine learning methods on tabular data.

**Method** We consider the tabular datasets from the Penn Machine Learning Benchmark (PMLB) from (Olson et al., 2017). These encapsulate a wide variety of real-world problems such as predicting chess moves, toxicity of mushrooms, credit scores and heart diseases. Since our model can only take binary features as input, we discard continuous features, and binarize the categorical features with $C > 2$ classes into $C$ binary variables. Note that this procedure can greatly increase the total number

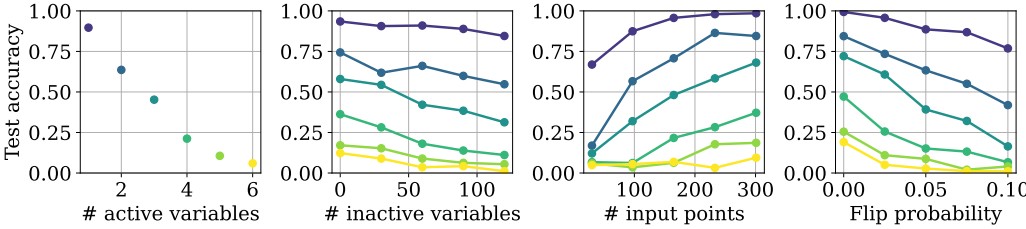

Figure 5: **Our model is robust to data incompleteness, bit flipping and noisy variables.** We display the test accuracy of our model when varying the four factors of difficulty described in Sec. 2. The colors depict different number of active variables, as shown in the first panel. Metrics are averaged over 10k samples from the random generator.

of features – we only keep datasets for which it results in less than 120 features (the maximum our model can handle). We randomly sample 25% of the examples for testing and report the F1 score obtained on this held out set.

We compare our model with two classic machine learning methods: logistic regression and random forests, using the default hyperparameters from `sklearn`. For random forests, we test two values for the number of estimators: 1 (in which case we obtain a simple decision tree as for the boolformer) and 100.

**Results** Results are reported in Fig. 6, where for readability we only display the datasets where the random forest with 100 estimators achieves an F1 score above 0.75. The performance of the Boolformer is similar on average to that of logistic regression: logistic regression typically performs better on "hard" datasets where there is no exact logical rule, for example medical diagnosis tasks such as `heart_h`, but worse on logic-based datasets where the data is not linearly separable such as `xd6`.

The F1 score of our model is slightly below that of a random forest of 100 trees, but slightly above that of the random forest with a single tree. This is remarkable considering that the Boolean formula it outputs only contains a few dozen nodes at most, whereas the trees of random forest use up to several hundreds. We show an example of a Boolean formula predicted for the mushroom toxicity dataset in Fig. 2, and a more extensive collection of formulas in App. E.

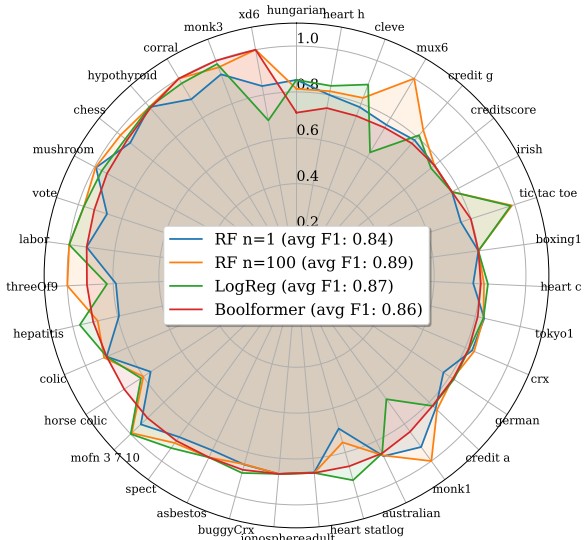

Figure 6: **Our model is competitive with classic machine learning methods while providing highly interpretable results.** We display the F1 score obtained on various binary classification datasets from the Penn Machine Learning Benchmark (Olson et al., 2017). We compare the F1 score of the Boolformer with random forests (using 1 and 100 estimators) and logistic regression, using the default settings of `sklearn`, and display the average F1 score of each method in the legend.

### 4.3 INFERRING BOOLEAN NETWORKS: APPLICATION TO GENE REGULATORY NETWORKS

A Boolean network is a dynamical system composed of $D$ bits whose transition from one state to the next is governed by a set of $D$ Boolean functions[7]. These types of networks have attracted a

---

[7]The $i$-th function $f_i$ takes as input the state of the $D$ bits at time $t$ and returns the state of the $i$-th bit at time $t+1$.

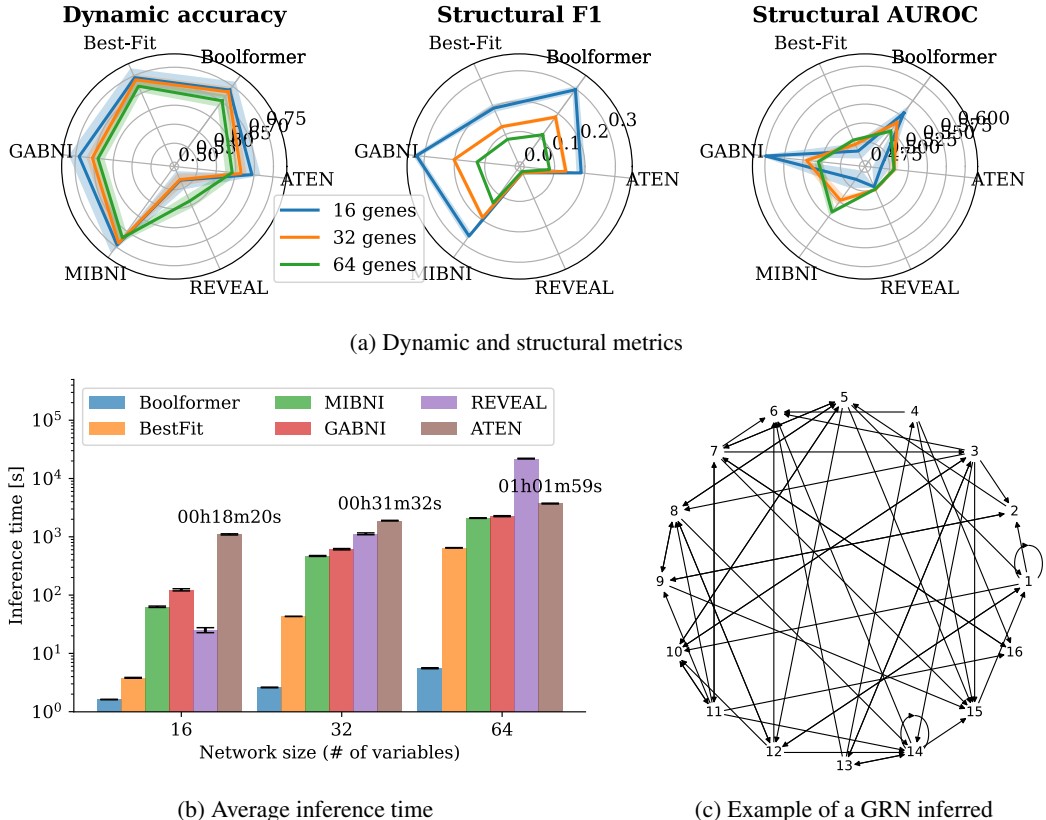

(a) Dynamic and structural metrics

(b) Average inference time

(c) Example of a GRN inferred

Figure 7: **Our model is competitive with state-of-the-art methods for GRN inference with orders of magnitude faster inference.** (a) We compare the ability of our model to predict the next states (dynamic accuracy) and the influence graph (structural accuracy) with that of other methods using a recent benchmark (Pušnik et al., 2022) – more details in Sec. 4.3. (b) Average inference time of the various methods. (c) From the Boolean formulas predicted, one can construct an influence graph where each node represents a gene, and each arrow signals that one gene regulates another.

lot of attention in the field of computational biology as they can be used to model gene regulatory networks (GRNs) (Zhao et al., 2021) – see App. F for a brief overview of this field. In this setting, each bit represents the (discretized) expression of a gene (on or off) and each function represents the regulation of a gene by the other genes. In this section, we investigate the applicability of our symbolic regression-based approach to this task.

**Benchmark** We use the recent benchmark for GRN inference introduced by Pušnik et al. (2022). This benchmark compares 5 methods for Boolean network inference on 30 Boolean networks inferred from biological data, with sizes ranging from 16 to 64 genes, and assesses both dynamical prediction (how well the model predicts the dynamics of the network) and structural prediction (how well the model predicts the Boolean functions compared to the ground truth). Structural prediction is framed as the binary classification task of predicting whether variable $i$ influences variable $j$, and can hence be evaluated by many binary classification metrics; we report here the structural F1 and the AUROC metrics which are the most holistic, and defer other metrics to App. F.

**Method** Our model predicts each component $f_i$ of the Boolean network independently, by taking as input the whole state of the network at times $[0 \ldots t-1]$ and as output the state of the $i$th bit at times $[1 \ldots t]$. Once each component has been predicted, we can build a causal influence graph, where an arrow connects node $i$ to node $j$ if $j$ appears in the update equation of $i$: an example is shown in Fig. 7c. Note that since the dynamics of the Boolean network tend to be slow, an easy way to get rather high dynamical accuracy would be to simply predict the trivial fixed point $f_i = x_i$. In

concurrent approaches, the function set explored excludes this solution; in our case, we simply mask the $i$th bit from the input when predicting $f_i$.

**Results** We display the results of our model on the benchmark in Fig. 7a. The Boolformer performs on par with the SOTA algorithms, GABNI (Barman & Kwon, 2018) and MIBNI (Barman & Kwon, 2017). A striking feature of our model is its inference speed, displayed in Fig. 7b: a few seconds, against up to an hour for concurrent approaches, which mainly rely on genetic programming. Note also that our model predicts an interpretable Boolean function, whereas the other SOTA methods (GABNI and MIBNI) simply pick out the most important variables and the sign of their influence.

## 5 DISCUSSION AND LIMITATIONS

In this work, we have shown that Transformers excel at symbolic regression of logical functions, both in the noiseless setup where they could potentially provide valuable insights for circuit design, and in the real-world setup of binary classification where they can provide interpretable solutions. Their ability to infer GRNs several orders of magnitude faster than existing methods offers the promise of many other exciting applications in biology, where Boolean modelling plays a key role (Hemedan et al., 2022). There are however several limitations in our current approach, which open directions for future work.

First, due to the quadratic cost of self-attention, the number of input points is limited to a thousand during training, which limits the model's performance on high-dimensional functions and large datasets (although the model does exhibit some length generalization abilities at inference, as shown in App. C). One could address this shortcoming with linear attention mechanisms (Choromanski et al., 2020; Wang et al., 2020), at the risk of degrading performance[8].

Second, the logical functions which our model is trained on do not include the XOR gate explicitly, limiting both the compactness of the formulas it predicts and its ability to express complex formulas such as parity functions. The reason for this limitation is that our generation procedure relies on expression simplification, which requires rewriting the XOR gate in terms of AND, OR and NOT. We leave it as a future work to adapt the generation of simplified formulas containing XOR gates, as well as operators with higher parity as in Rosenberg et al. (2023).

Third, the simplicity of the formulas predicted is limited in two additional ways: our model only handles (i) single-output functions – multi-output functions are predicted independently component-wise and (ii) gates with a fan-out of one[9]. As a result, our model cannot reuse intermediary results for different outputs or for different computations within a single output[10]. One could address this either by post-processing the generated formulas to identify repeated substructures, or by adapting the data generation process to support multi-output functions (a rather easy extension) and cyclic graphs (which would require more work).

Finally, this paper mainly focused on investigating concrete applications and benchmarks to motivate the potential and development of Boolformers. However, the ability to control the formula generator can help correcting the simplicity bias of the model in order to help with reasoning tasks. In future research, we will tackle various theoretical aspects of this paradigm, playing with the model simplicity bias to investigate the sample complexity and 'generalization on the unseen' (Abbe et al., 2023) of the Boolformer.

We conclude by emphasizing that the difficulty and potential ambiguity in inferring logical functions calls for caution during deployment. Methods like the Boolformer primarily serve as hypothesis generators, ultimately requiring further experimental verification.

---

[8]We hypothesize that full attention span is particularly important in this specific task: the attention maps displayed in App. H are visually quite dense and high-rank matrices.

[9]Note that although the fan-in is fixed to 2 during training, it is easy to transform the predictions to larger fan-in by merging ORs and ANDs together.

[10]Consider the $D$-parity: one can build a formula with only $3(n-1)$ binary AND-OR gates by storing $D-1$ intermediary results: $a_1 = XOR(x_1, x_2), a_2 = XOR(a_1, x_3), \ldots, a_{n-1} = XOR(a_{D-2}, x_D)$. Our model needs to recompute these intermediary values, leading to much larger formulas, e.g. 35 binary gates instead of 9 for the 4-parity as illustrated in App. D.

## REPRODUCIBILITY STATEMENT

The reproducibility of our work is ensured through several means. All code and model weights will be made publicly available at `anonymized` together with notebooks to directly reproduce key aspects of the results, as well as a pip-installable package for easy usage. We also describe in detail the data generation, our architecture and training choices in Sec. 2.

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
