## A  EXPRESSION SIMPLIFICATION

The data generation procedure heavily relies on expression simplification. This is of utmost importance for four reasons:

- It reduces the output expression length and hence memory usage as well as increasing speed
- It improves the supervision by reducing expressions to a more canonical form, easier to guess for the model
- It increases the effective diversity of the beam search, by reducing the number of equivalent expressions generated
- It encourages the model to output the simplest formula, which is a desirable property.

We use the package `boolean.py`[11] for this, which is considerably faster than `sympy` (the function `simplify_logic` of the latter has exponential complexity, and is hence only implemented for functions with less then 9 input variables).

Empirically, we found the following procedure to be optimal in terms of average length obtained after simplification:

1. Preprocess the formula by applying basic logical equivalences: double negation elimination and De Morgan's laws.
2. Parse the formula with `boolean.py` and run the `simplify()` method *twice*
3. Apply once again the first step

Note that this procedure drastically reduces the number of operators and renders the final distribution highly nonuniform, as shown in Fig. 8.

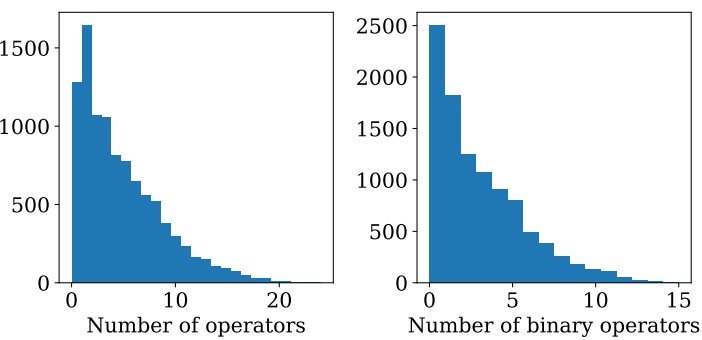

Figure 8: Distribution of number of operators after expression simplification. The initial number of binary operators is sampled uniformly in $[1, 1000]$. The total number of examples is $10^4$.

## B  DOES THE BOOLFORMER MEMORIZE?

One natural question is whether our model simply performs memorization on the training set. Indeed, the number of possible functions of $D$ variables is finite, and equal to $2^{2^D}$.

Let us first assume naively that our generator is uniform in the space of boolean functions. Since $2^{2^4} \simeq 6 \times 10^4$ (which is smaller than the number of examples seen during training) and $2^{2^5} \simeq 5.10^9$ (which is much larger), one could conclude that for $D \le 4$, all functions are memorized, whereas for $D > 4$, only a small subset of all possible functions are seen, hence memorization cannot occur.

However, the effective number of unique functions seen during training is actually smaller because our generator of random functions is nonuniform in the space of boolean functions. In this case, for which value of $D$ does memorization become impossible? To investigate this question, for each

---

[11] https://github.com/bastikr/Boolean.py

$D < D_{\max}$, we sample $\min\left(2^{2^D}, 100\right)$ unique functions from our random generator, and count how many times their exact truth table is encountered over an epoch (300,000 examples).

Results are displayed in Fig. 9. As expected, the average number of occurences of each function decays exponentially fast, and falls to zero for $D = 7$, meaning that each function is typically unique for $D \geq 7$. Hence, memorization cannot occur for $D \geq 7$. Yet, as shown in Fig. 4, our model achieves excellent accuracies even for functions of 10 variables, which excludes memorization as a possible explanation for the ability of our model to predict logical formulas accurately.

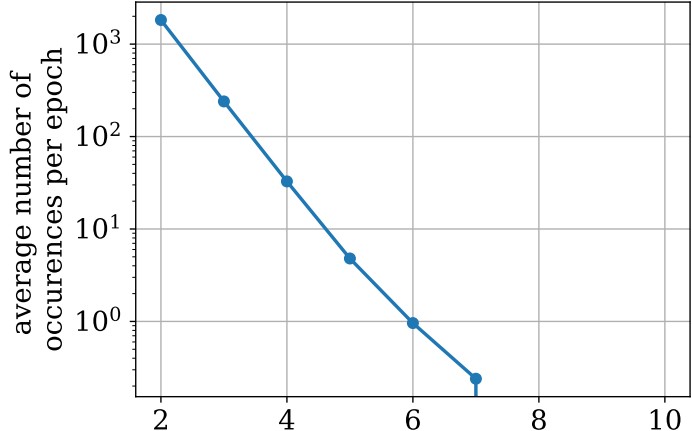

Figure 9: **Functions with 7 or more variables are typically never seen more than once during training.** We display the average number of times functions of various input dimensionalities are seen during an epoch (300,000 examples). For each point on the curve, the average is taken over $\min(2^{2^D}), 100)$ unique functions.

## C  LENGTH GENERALIZATION

In this section we examine the ability of our model to length generalize. In this setting, there are two types of generalization one can define: generalization in terms of the number of inputs $N$, or in terms of the number of active variables $S$[12]. We examine length generalization in the noisy setup (see Sec. 2.2), because in the noiseless setup, the model already has access to all the truth table (increasing $N$ does not bring any extra information), and all the variables are active (we cannot increase $S$ as it is already equal to $D$).

### C.1  NUMBER OF INPUTS

Since the input points fed to the model are permutation invariant, our model does not use any positional embeddings. Hence, not only can our model handle $N > N_{\max}$, but performance often continues to improve beyond $N_{\max}$, as we show for two datasets extracted from PMLB (Olson et al., 2017) in Fig. 10.

### C.2  NUMBER OF VARIABLES

To assess whether our model can infer functions which contain more active variables than seen during training, we evaluated a model trained on functions with up to 6 active variables on functions with 7 or more active variables. We provided the model with the truth table of two very simply functions: the OR and AND of the first $S \geq 7$ variables. We observe that the model succeeds for $S = 7$, but fails for $S \geq 8$, where it only includes the first 7 variables in the OR / AND. Hence, the model can length generalize to a small extent in terms of number of active variables, but less easily than in

---

[12]Note that our model cannot generalize to a problem of higher dimensionality $D$ than seen during training, as its vocabulary only contains the names of variables ranging from $x_1$ to $x_{D_{\max}}$.

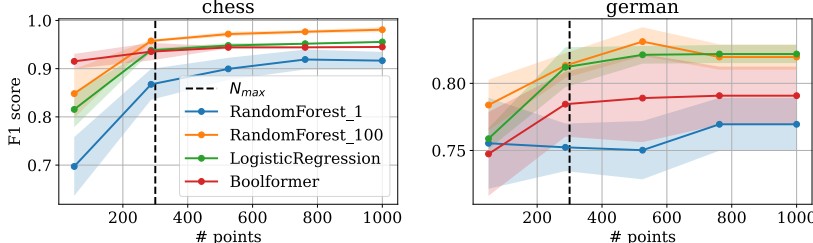

Figure 10: **Our model can length generalize in terms of sequence length.** We test a model trained with $N_{\max} = 300$ on the chess and german datasets of PMLB. Results are averaged over 10 random samplings of the input points, with the shaded areas depicting the standard deviation.

terms of number of inputs. We hypothesize that proper length generalization could be achieved by "priming", i.e. adding even a small number of "long" examples, as performed in Jelassi et al. (2023).

## D FORMULAS PREDICTED FOR LOGICAL CIRCUITS

In Figs. 11 and 12, we show examples of some common arithmetic and logical formulas predicted by our model in the noiseless regime, with a beam size of 100. In all cases, we increase the dimensionality of the problem until the failure point of the Boolformer.

## E FORMULAS PREDICTED FOR PMLB DATASETS

In Fig. 13, we report a few examples of boolean formulas predicted for the PMLB datasets in Fig. 6. In each case, we also report the F1 scores of logistic regression and random forests with 100 estimators.

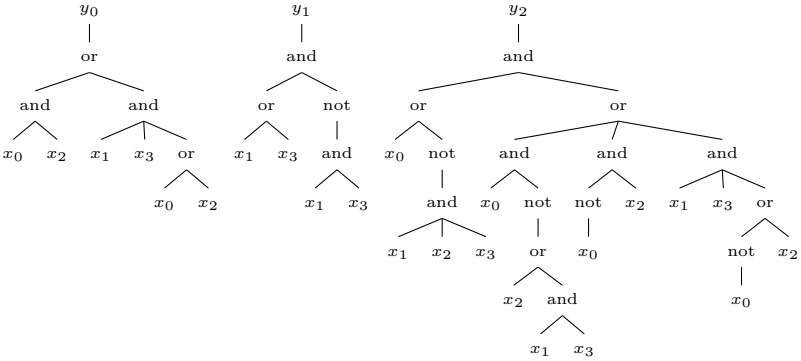

(a) Addition of two 2-bit numbers: $y_0 y_1 y_2 = (x_0 x_1) + (x_2 x_3)$. All formulas are correct.

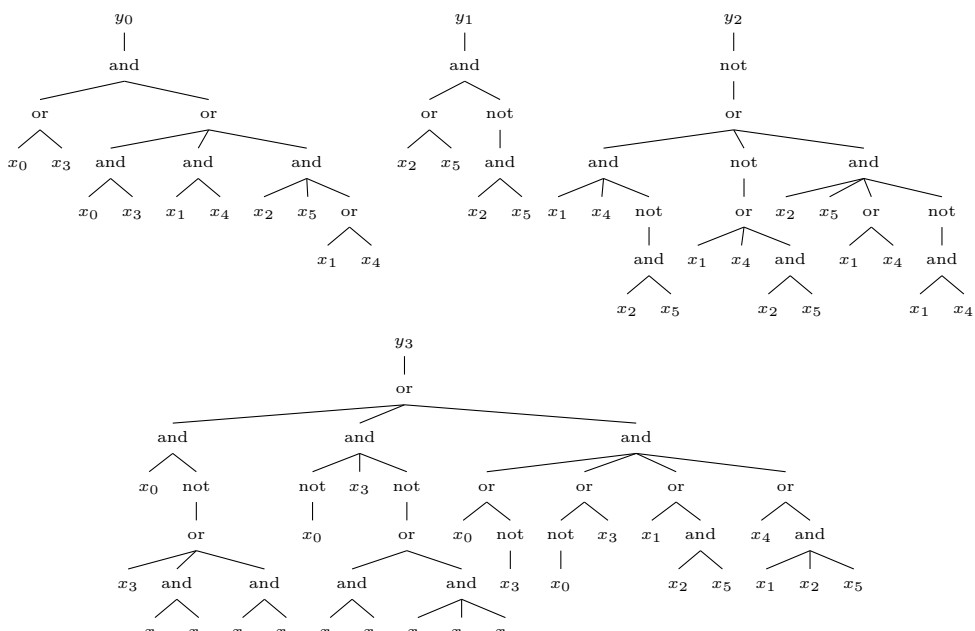

(b) Addition of two 3-bit numbers: $y_0 y_1 y_2 y_3 = (x_0 x_1 x_2) + (x_3 x_4 x_5)$. All formulas are correct, except $y_3$ which gets an error of 3%.

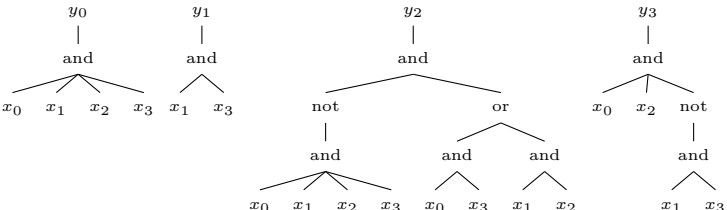

(c) Multiplication of two 2-bit numbers: $y_0 y_1 y_2 y_3 = (x_0 x_1) \times (x_2 x_3)$. All formulas are correct.

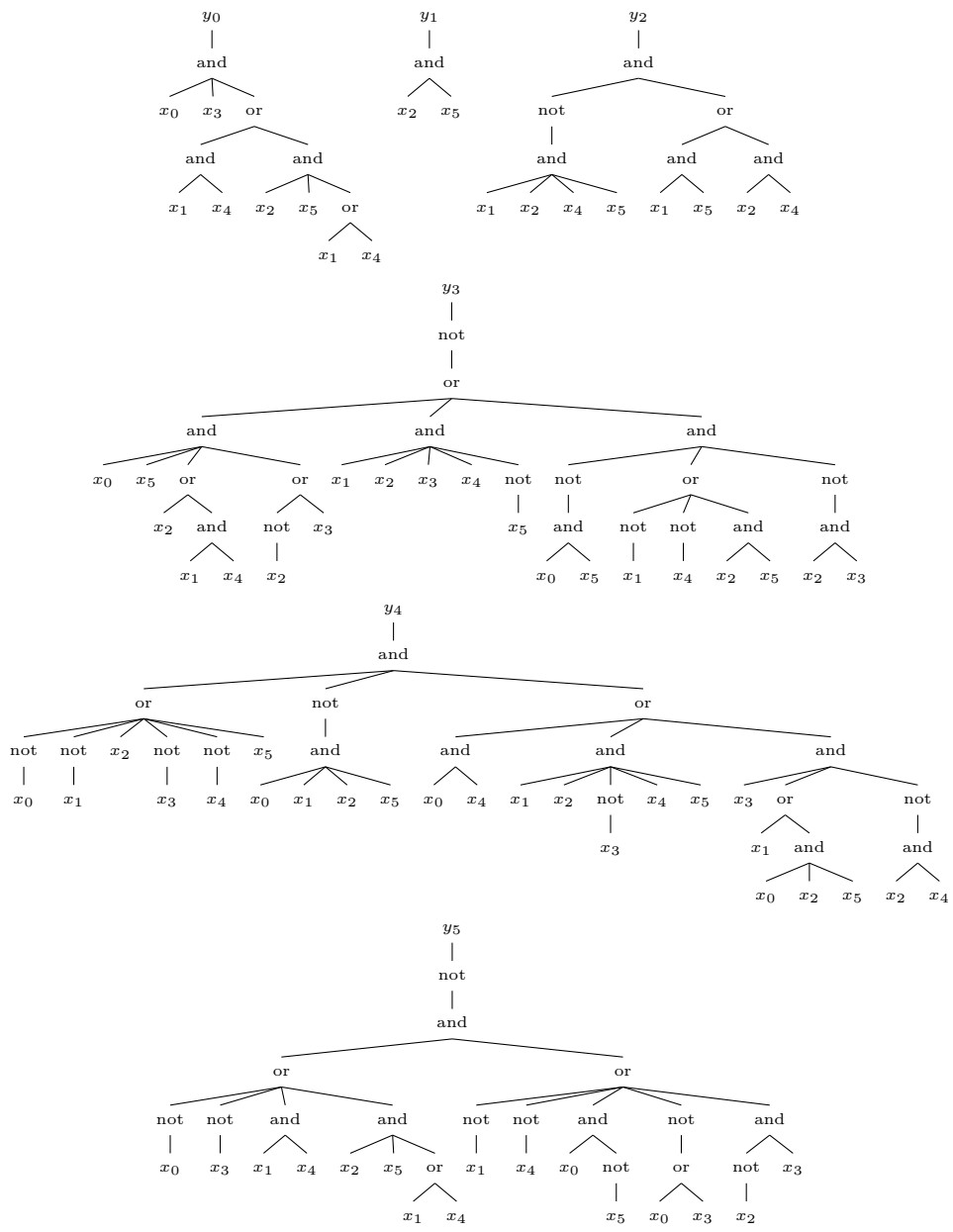

(d) Multiplication of two 3-bit numbers: $y_0 y_1 y_2 y_3 y_4 y_5 = (x_0 x_1 x_2) \times (x_3 x_4 x_5)$. All formulas are correct, except $y_4$ which gets an error of 5%.

Figure 11: **Some arithmetic formulas predicted by our model.**

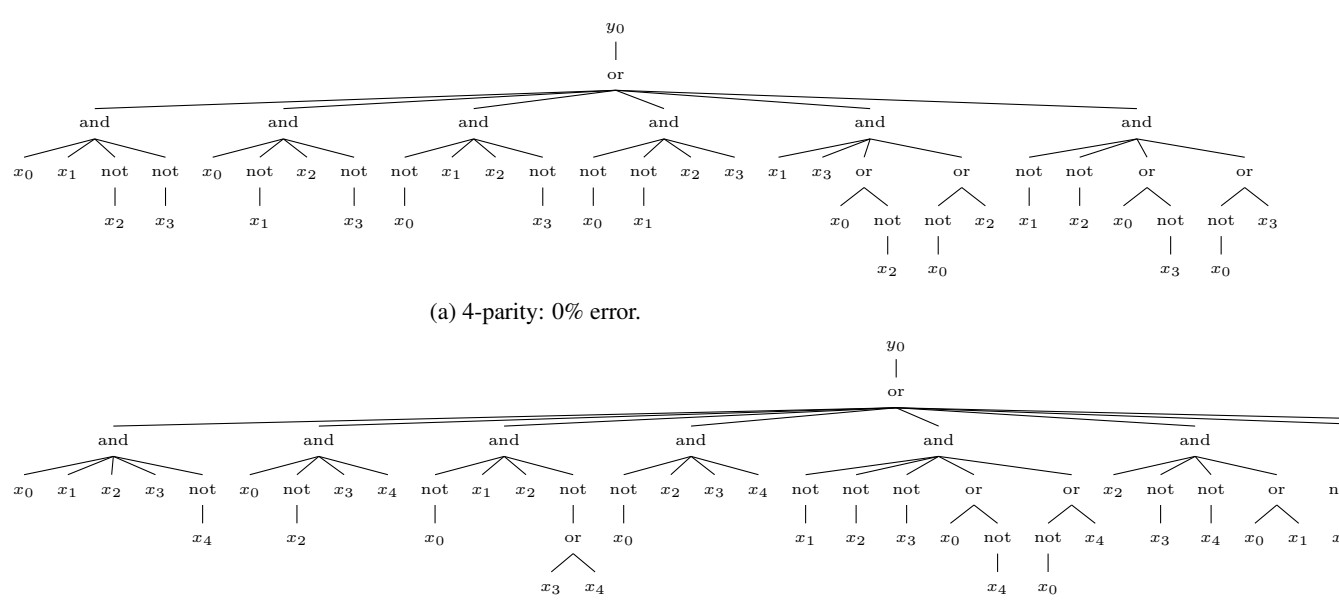

(a) 4-parity: 0% error.

(b) 5-parity: 28% error.

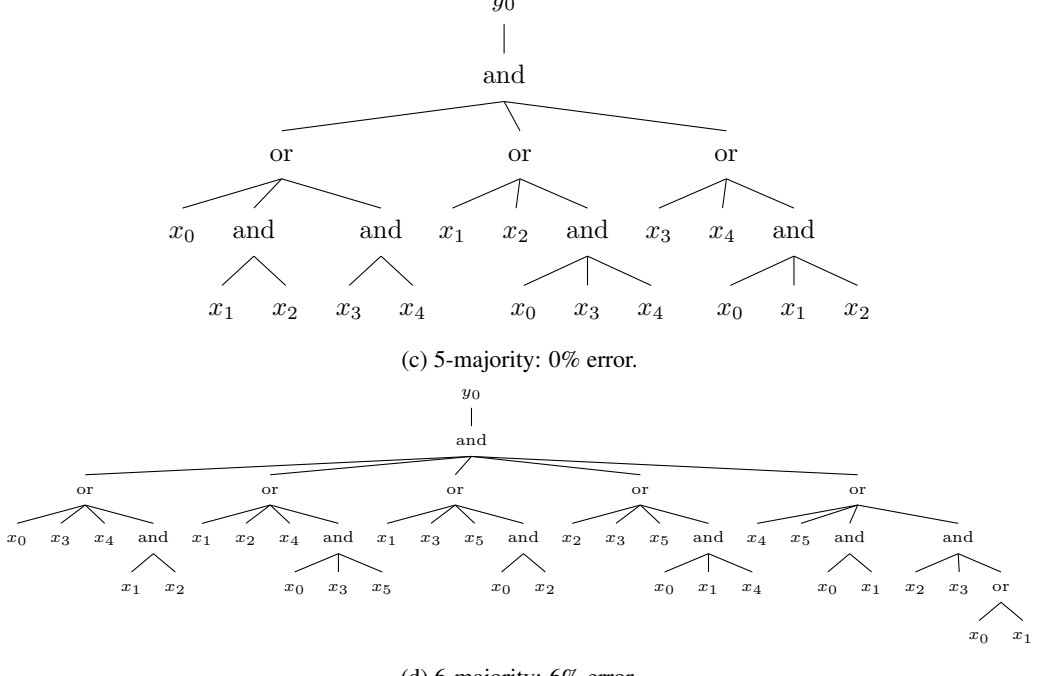

(c) 5-majority: 0% error.

(d) 6-majority: 6% error.

Figure 12: **Some logical functions predicted by our model.**

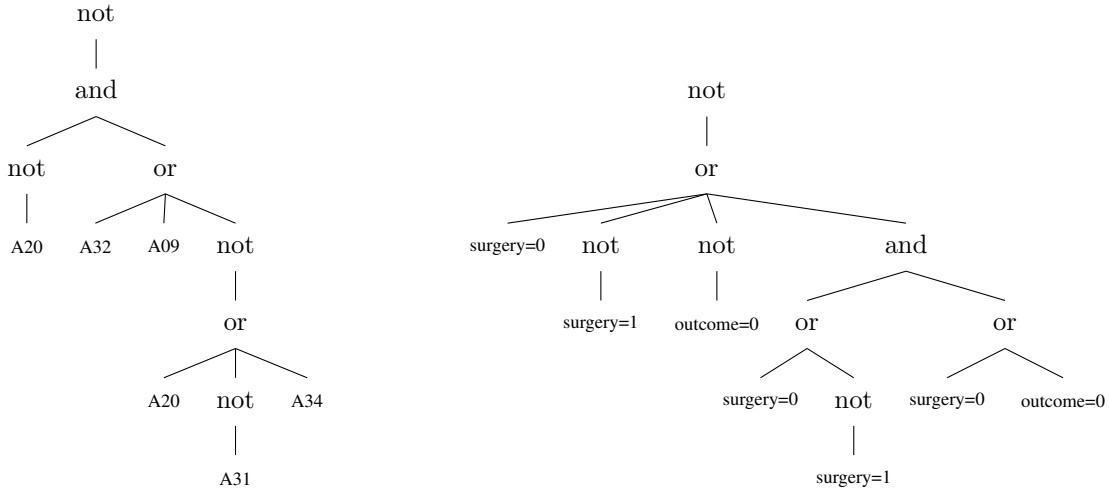

(a) chess. F1: 0.947. LogReg: 0.958. RF: 0.987.  (b) horse colic. F1: 0.900. LogReg: 0.822. RF: 0.861.

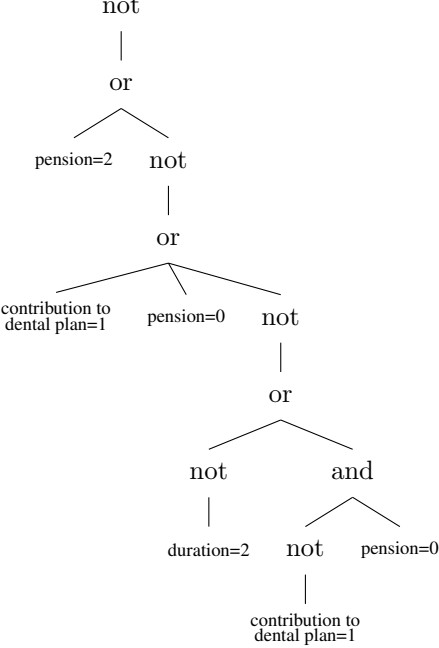

(c) labor. F1: 0.960. LogReg: 1.000. RF: 1.000.

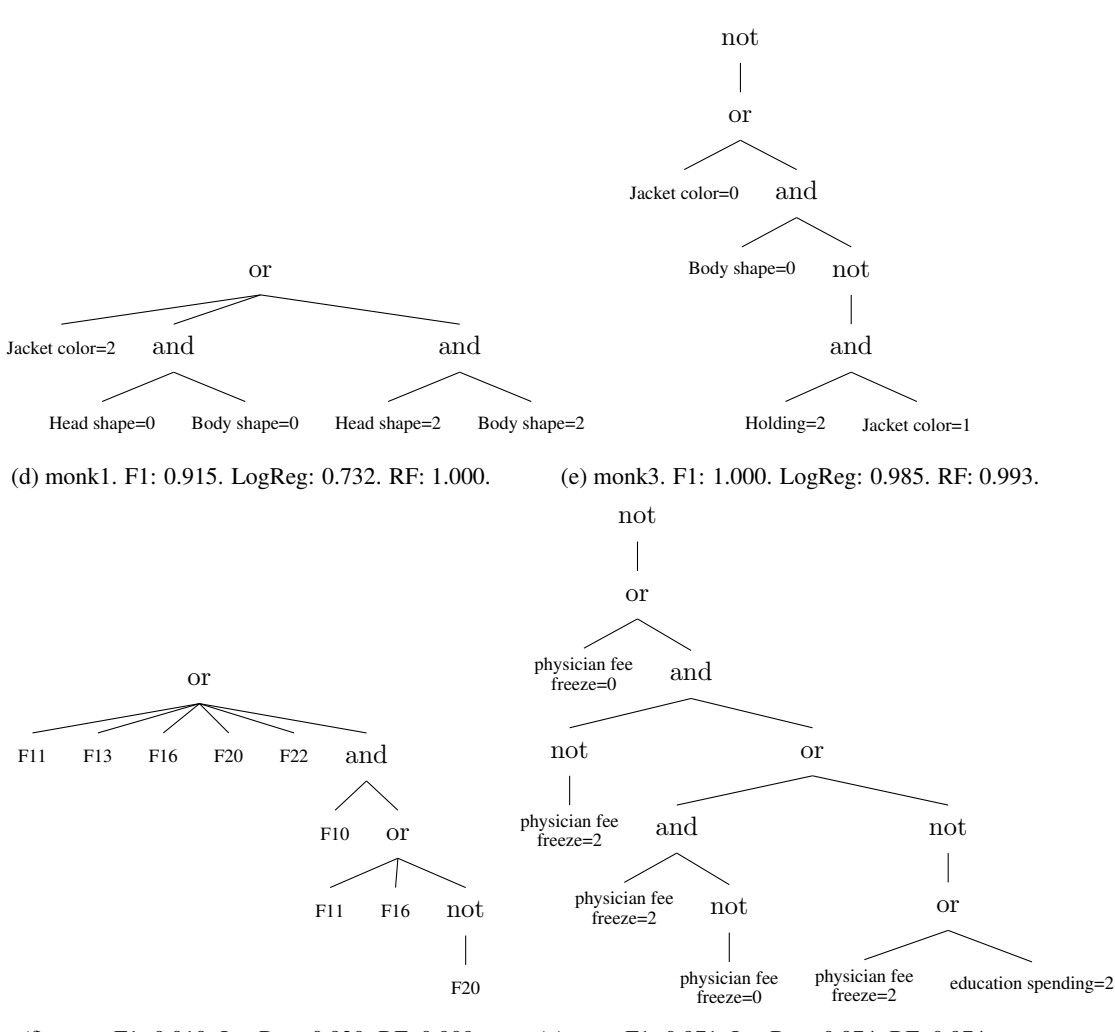

(d) monk1. F1: 0.915. LogReg: 0.732. RF: 1.000.

(e) monk3. F1: 1.000. LogReg: 0.985. RF: 0.993.

(f) spect. F1: 0.919. LogReg: 0.930. RF: 0.909.

(g) vote. F1: 0.971. LogReg: 0.974. RF: 0.974.

Figure 13: **Some logical formulas predicted by our noisy model for some binary classification PMLB datasets.** In each case, we report the name of the dataset and the F1 score of the Boolformer, logistic regression and random forest in the caption.

# F  ADDITIONAL RESULTS ON GENE REGULATORY NETWORK INFERENCE

In this section, we give an brief overview of the field of GRN inference and present additional results using our Boolformer.

## F.1  A BRIEF OVERVIEW OF GRNS

Inferring the behavior of GRNs is a central problem in computational biology, which consists in deciphering the activation or inhibition of one gene by another gene from a set of noisy observations. This task is very challenging due to the low signal-to-noise ratios recorded in biological systems, and the difficulty to obtain temporal ordering and ground truth networks.

GRN algorithms can infer relationships between the genes based on static observations Singh & Vidyasagar (2015); Haury et al. (2012); Huynh-Thu et al. (2010), or on input time-series recordings Adabor & Acquaah-Mensah (2019); Huynh-Thu & Geurts (2018), and can either infer correlational relationships, i.e. undirected graphs, or causal relationships, i.e. directed graphs – the latter being more useful, but harder to obtain.

We focus on methods which model the dynamics of GRNs via Boolean networks: REVEAL (Liang et al., 1998), Best-Fit (Lähdesmäki et al., 2003), MIBNI (Barman & Kwon, 2017), GABNI (Barman & Kwon, 2018) and ATEN (Shi et al., 2020a). We evaluate our approach on the recent benchmark from Pušnik et al. (2022).

## F.2  ADDITIONAL RESULTS

The benchmark studied in the main text assesses both dynamical prediction (how well the model predicts the dynamics of the network) and structural prediction (how well the model predicts the Boolean functions compared to the ground truth). Structural prediction is framed as the binary classification task of predicting whether variable $i$ influences variable $j$, and can hence be evaluated by several binary classification metrics, defined below[13]:

$$\text{Acc} = \frac{\text{TP} + \text{TN}}{\text{TP} + \text{TN} + \text{FP} + \text{FN}}, \quad \text{Pre} = \frac{\text{TP}}{\text{TP} + \text{FP}}, \quad \text{Rec} = \frac{\text{TP}}{\text{TP} + \text{FN}}, \quad F1 = 2\frac{\text{Pre} \cdot \text{Rec}}{\text{Pre} + \text{Rec}},$$

$$\text{MCC} = \frac{\text{TP} \cdot \text{TN} - \text{FP} \cdot \text{FN}}{\sqrt{(\text{TP} + \text{FP})(\text{TP} + \text{FN})(\text{TN} + \text{FP})(\text{TN} + \text{FN})}}, \quad \text{AUROC} = \frac{\text{TP}}{\text{TP} + \text{FN}} + \frac{\text{TN}}{\text{TN} + \text{FP}} - 1.$$

We report these metrics in Fig. 14.

---

[13]The authors of the benchmark consider the two latter to be the best metrics to give a comprehensive view on the classifier performance for this task.

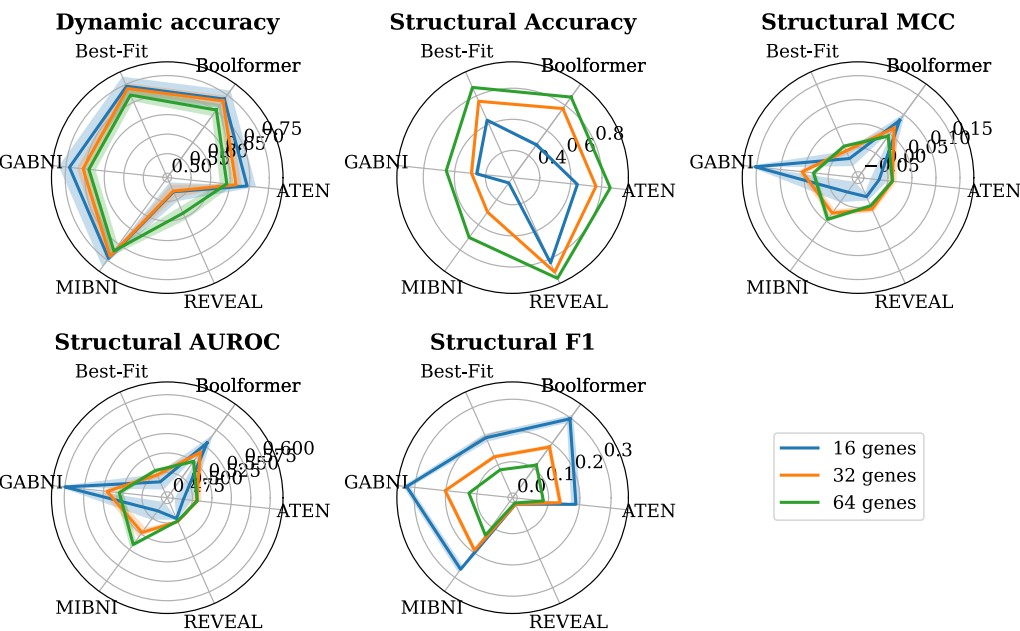

Figure 14: **Binary classification metrics used in the gene regulatory network benchmark.** The competitors and metrics are taken from the recent benchmark of Pušnik et al. (2022), and described in Sec. 4.3.

## G  EXPLORING THE BEAM CANDIDATES

In this section, we explore the beam candidates produced by the Boolformer. In Fig. 15, we show the 8 top-ranked candidates when predicting a simple logic function, the 2-comparator. We see that all candidates perfectly match the ground truth, but have different structure.

## H  ATTENTION MAPS

In Fig. 16, we show the attention maps produced by our model when presented three truth tables: (a) that of the 4-digit multiplier, (b) that of the 4-parity function and (c) a random truth table. Each panel corresponds to a different layer and head of the model.

Recall that each of the $N$ inputs to the transformer is the embedding of an $(\mathbf{x}, y)$ pair; in this case, we ordered the embeddings according to the binary number they form, i.e. from left to right: 0000, 0001, 0010, ..., 1111. We see highly structured patterns emerging, especially for the first two functions which are non-random.

For example, for the 4-digit multiplier, some attention heads have hadamard-like structure (e.g. head 5 of layer 6), some have block-structured checkboard patterns (e.g. head 12 of layer 4), and many heads put most attention weight on the final input, 1111, which is more informative (e.g. head 6 of layer 3).

For the parity function, we see a particularly interesting shape emerge in several heads (e.g. head 2 of layer 4), and observe that many heads perform anti-diagonal attention (e.g. head 4 of layer 6).

## I  EMBEDDINGS

In this section, we show that the model learns a compressed representation of the hypercube which conserves distances, both qualitatively in Fig. 17, and quantitatively in Fig. 18, where we plot the L2 distance in embedding space against the hamming distance in input space, showing a linear relationship.

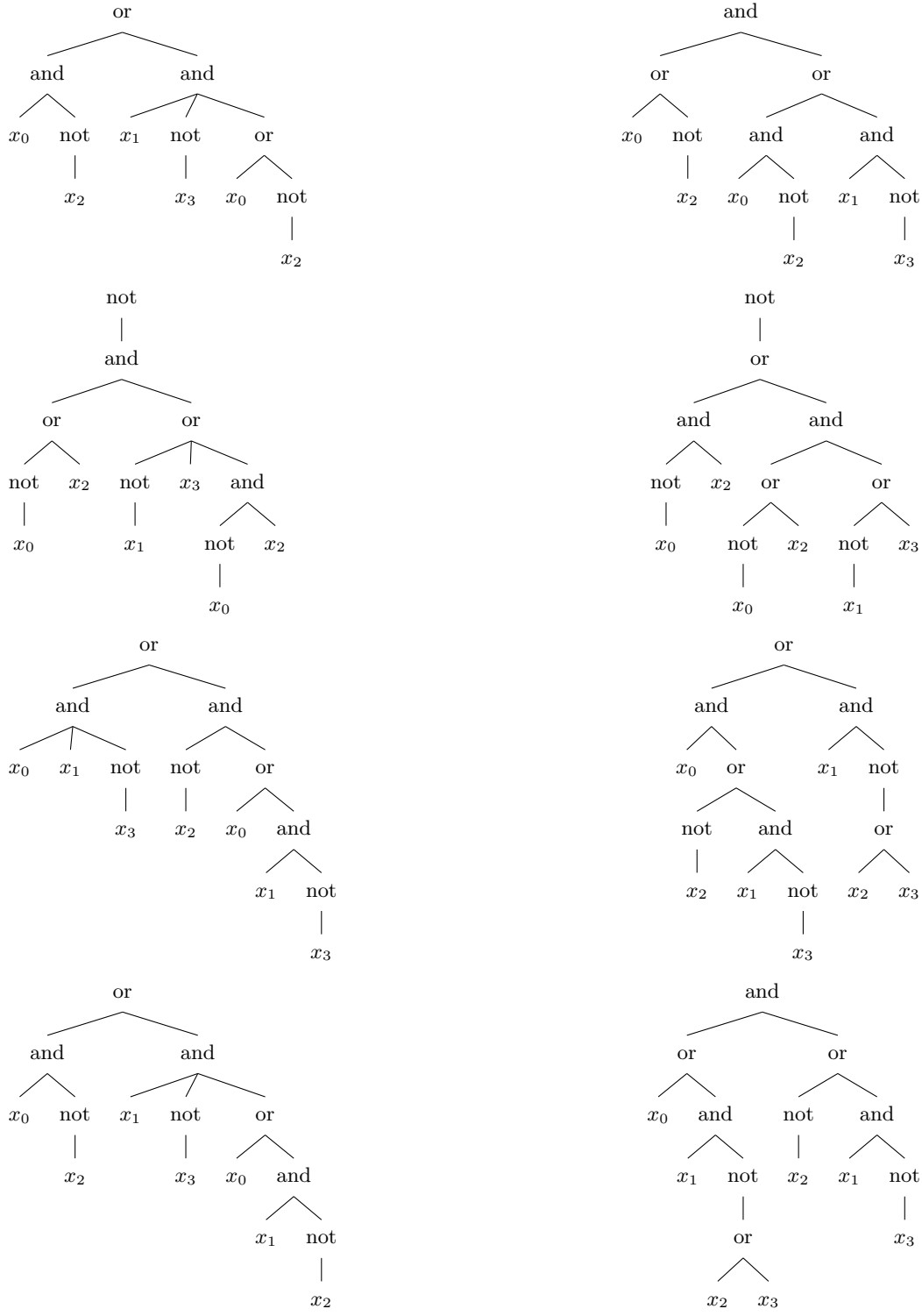

Figure 15: **Beam search reveals equivalent formulas**. We show the first 8 beam candidates for the 2-comparator, which given two 2-bit numbers $a = (x_0 x_1)$ and $b = (x_2 x_3)$, returns 1 if $a > b$, 0 otherwise. All candidates perfectly match the ground truth.

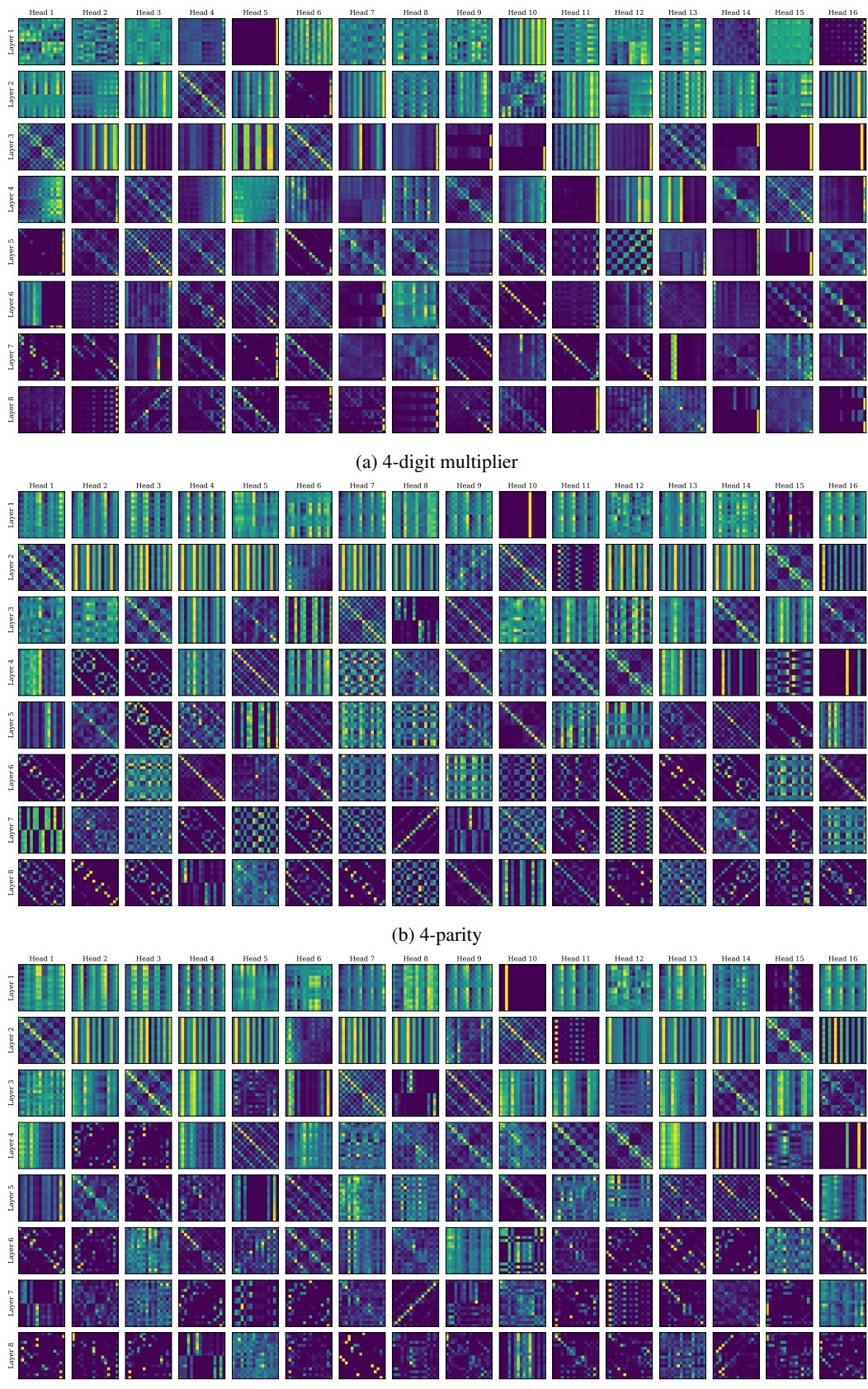

(a) 4-digit multiplier

(b) 4-parity

(c) 4d random data

Figure 16: **The attention maps reveal intricate analysis.** See Sec. H for more details on this figure.

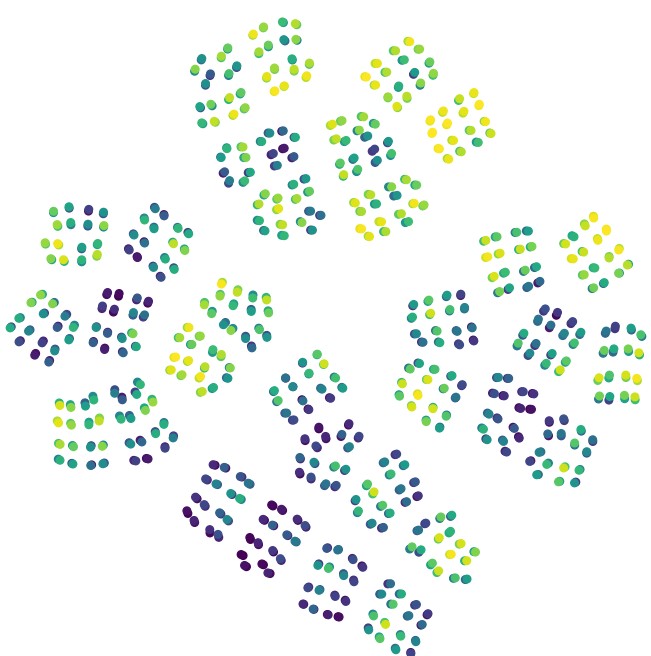

Figure 17: **T-SNE representation of the embeddings.** We fed the 1024 input combinations of the 10-dimensional hypercube to the embedder, and colored them according to the number of 1's, from 0 (blue, which corresponds to 0000000000) to 10 (yellow, which corresponds to 1111111111).

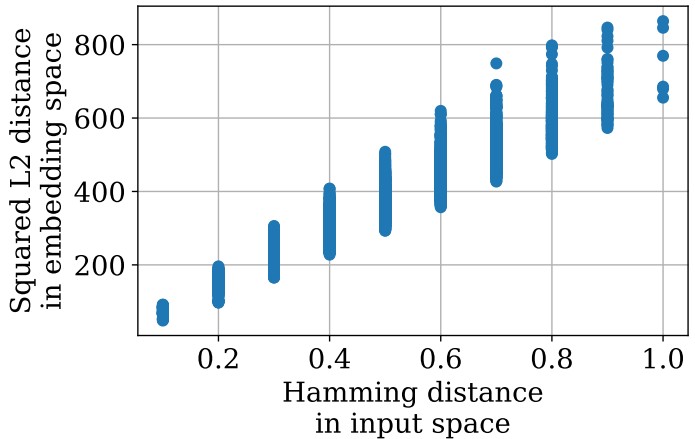

Figure 18: **The embedder conserves distances.** We plot the squared L2 distance of the embeddings of all points in the 10-dimensional hypercube against the hamming distance in the input space.