# OpenReview forum: "Boolformer: Symbolic Regression of Logic Functions with Transformers"
_ICLR.cc/2024/Conference — Submitted to ICLR 2024_

### Official Review · Reviewer_V2Ca · 2023-10-31

**Soundness:** 2 fair
**Presentation:** 3 good
**Contribution:** 2 fair
**Rating:** 3
**Confidence:** 3

**Summary:**

The paper proposes a model to perform symbolic regression of boolean functions using transformers

**Strengths:**

The method is sound, simple and elegant.

Boolean functions are easy to be generated randomly in huge amount, which is where transformers architecture shine.
The  techniques is straightforward: generate boolean formulas randomly (covering the space of function as much as possible with a bias towards short formulas) and train a seq-2-seq architecture on that.

The speed up on gene-regulatory networks is interesting.

**Weaknesses:**

The major weakness of the paper is the positioning w.r.t. the state of the art. Being the method very simple, clearly highlighting the novelty should have been a priority.

While the paper discusses many related areas, it is very unclear what is new in the proposed approach. For example, the section on "symbolic regression" in the related work, which is the closest area to the proposed approach ("Symbolic regression of logic functions"), is simply a list of papers. The approach is not compared with these approaches neither experimentally not even theoretically.

Experiments in the noiseless regime do not compare Boolformer with any baseline (therefore is quite hard to understand how hard is the task overall).

Experiments in the noisy regime have comparisons but with very unrelated approaches (generic ML models or specific to the dataset)

Minor: The numbers in the radar charts in Figure 7 are impossible to read.

**Questions:**

1) Would be possible to apply any existing symbolic regression approaches to the proposed task?

2) How novel is the generation of boolean formulas? Are there similar ideas in the literature to generate datasets for symbolic regression?

3) How can you measure how hard is the task? Would any other method (both transformer based, or tradition ILP setting, be able to solve the task to some extent?

---

> ### Author Response · Authors · 2023-11-19
>
> We thank the reviewer for their constructive feedback.
>
> Concerning the novelty of this paper and comparisons, we refer to our introductory comments.
> Similarly, the idea of generating synthetic datasets has been explored for symbolic regression of real-valued functions, but never in the Boolean setting.
>
> We would also like to emphasize that the problem studied in the noiseless setting, also known as the Minimum Circuit Size Problem, is an NP-hard problem which no existing methods can currently solve, including ours. This section is more theoretical in nature, and was precisely designed to probe the ability of our model to generalize (i.e., prove that memorization does not occur, as discussed more in App. B) as well as its limitations of our model – we will clarify this in the revised manuscript.
>
> Crucially, the core contribution of this work in terms of practical relevance lies in the noisy setting, where we provide extensive comparisons.
> See more details in the introductory comments.

---

> > ### Comment · Reviewer_V2Ca · 2023-11-23
> >
> > I thank the reviewers for their answer. However, as I stated in my original review, I have the same feeling as reviewer BZRx that the novelty of the method is extremely limited. Except for the dataset, all the technicalities are taken from other few papers (with almost no modification). I continue to believe that the paper is nice but, at this stage, the novelty is too limited for a ICLR publication.

---

### Official Review · Reviewer_BZRx · 2023-10-31

**Soundness:** 3 good
**Presentation:** 2 fair
**Contribution:** 2 fair
**Rating:** 5
**Confidence:** 4

**Summary:**

This paper proposes a method named Boolformer that performs symbolic regression for logical operations using AND, OR, and NOT. Boolformer is trained to output logic formulas in Polish notation using a Transformer-based Encoder-Decoder model. The results show not only the evaluation of the automatically generated formulas, but also the inference performance, excellent speed, and high explanatory power on PMLB databases and gene regulartory networks (GRNs) as real-world problems.

**Strengths:**

- A new problem setting in which logical expressions are symbolically regressed by Transformer.
- Experimental results on several real-world applications as well as on the generated logical equation data are reported.
  - The results using PMLB database shows accuracy comparable to Random Forest and logistic regression, and furthermore, the learned models are expected to provide excellent explanatory properties.
  - Using GRNs, Boolformer is shown to have both excellent accuracy and speed.

**Weaknesses:**

- The problem statement in the introduction does not match the solution. In Section 1, citing (Abbe et al., 2022), the authors point out that Transformer learns complex models in terms of the Fourier spectrum, resulting in poor generalization performance when samples are insufficient. As a contribution, Section 1.1 claims that it is robust to noisy and incomplete observations. However, the Boolformer proposed in this paper is a relatively natural application of the Transformer, and there is no redesign from a Fourier spectrum perspective or other robustness innovations.
- Due to the design of the method, it can only accept datasets with relatively few variables or small scale. This is acknowledged by the authors in section 5, but since there are currently proposals such as Transformer that can accept long series, it would have been easy to consider improving the limitation.
- For example, if the correct answer is [AND, X_1, NOT, X_2], then [AND, NOT, X_2, X_1] is also equivalent. The fact that the system is learned by cross-entropy means that it is unclear how a valid cross-entropy can be calculated when there are multiple correct answers in this way.

**Questions:**

- The reviewer expects the authors to respond to the points listed in Weaknesses.
- In the radar charts in Figure 7(a), the different methods are plotted among different axes, making it difficult to understand the comparison between those methods. If the radar chart is used to make comparisons between methods, it would be better to have as many axes as the number of experimental settings, such as the number of genes, and plot entities for each method. Alternatively, a table or a bar chart like Figure 7(b) is easier to compare methods.
- Minor comments:
  - In the caption of Figure 1, (x_5 x_6 x_7 x_7 x_9( should be (x_5 x_6 x_7 x_**8** x_9).
  - References should be corrected. Especially, many published papers are cited as preprints. Below are some examples:
    - The reference for (Abbe et al., 2022) should be a NeurIPS 2022 paper.
    - (Dosovitskiy et al., 2020) should be an ICLR 2021 paper.

---

> ### Author Response · Authors · 2023-11-19
>
> Weaknesses
>
> W1: We would like to defer the reviewer to the general comment, as this important point was also brought up by another reviewer.
>
> W2: Thank you for bringing up this important question, which indeed deserves clarifications. To extend our discussion on this: even if we were to use a linear attention method (which would likely be very detrimental as the attention maps shown in the appendix are particularly non-local and dense), the scaling would be nontrivial as the number of input points remains in the worst-case exponential in D. More generally, the design choice for improving long context prediction is not critical to our contribution since our contribution is intended to show the promise of symbolic regression for boolean circuits rather than to exhaustively search the right architecture for scaling.
>
> W3: This is an interesting point which we will also discuss more. In fact, it turns out that the fact that multiple expressions can evaluate to the same function is not a problem in our setup, and can in fact be beneficial as a form of data augmentation. Indeed, our model learns these invariances, as evidenced by Appendix G, which shows the candidates obtained via beam search : the first eight candidates are all valid formulas, written in different ways (and in all cases, very compact).
> In the context of symbolic regression, this aligns with the observation made in the Appendix of [1] that removing this equivalence via deterministic simplification rules does not yield any performance improvements.
> The fact that cross-entropy can handle expressions with multiple possible formulations is also observed in many other fields, e.g. in machine translation, where beam search also reveals the equivalent translations of a given sentence.
>
> Questions:
> We thank the reviewer for these very valuable remarks. For the radar plots, we reused the formatting of the authors of the benchmark [2] for maximal consistency, but will also add some bar charts.
>
> [1] d’Ascoli et al., Deep Symbolic Regression for Recurrent Sequences, ICML 2021
>
> [2] Žiga Pušnik, Miha Mraz, Nikolaj Zimic, and Miha Moškon. Review and assessment of boolean approaches for inference of gene regulatory networks. Heliyon, pp. e10222, 2022.

---

> > ### Comment · Reviewer_BZRx · 2023-11-20
> > **The rating remains the same**
> >
> > W1: The reviewer read the general comment. But it does not appear to be a response to the reviewer's point about the inconsistency between the problem in the introduction and the solution. After all, the robustness of the proposed Boolformer seems to be what the original Transformer has.
> >
> > W2: Related to the previous weakness, the technical contributions are still limited. The authors say it is a PoC, but it is not clear whether the concept should be taken up by ICLR.
> >
> > W3: In [1], it is shown that normalizing the equation using SymPy has no effect. This is only one option and does not indicate that loss function considerations for the diversity of the equation are unnecessary.
> >
> > Q: The radar chart in [2] can be used to comapre among the network node numbers, but it is difficult to compare among methods. The authors should consider more appropriate visualization methods; the visualization method adopted in one past paper is not a de-facto standard as it is.

---

> > > ### Author Response · Authors · 2023-11-20
> > > **more detailed response and corrections on points raised by the reviewer**
> > >
> > > W1-W2: We expand our responses to clarify - sorry if the previous responses were limited.
> > > The reviewer's comment "In Section 1, citing (Abbe et al., 2022), the authors point out that Transformer learns complex models in terms of the Fourier spectrum, resulting in poor generalization performance when samples are insufficient. As a contribution, Section 1.1 claims that it is robust to noisy and incomplete observations. However, the Boolformer proposed in this paper is a relatively natural application of the Transformer, and there is no redesign from a Fourier spectrum perspective or other robustness innovations." is not quite an accurate re-transcription of the paper's referencing to Abbe et al. nor the current paper's contributions. Let us try to clarify these:
> > >
> > > 1. Abbe et al. shows (experimentally for Transformers and provably for simpler models) that if you train a classic Transformer on a sparse Boolean function, with incomplete data in the sense that you do not sample part of the distribution domain (unseen domain) while you sample heavily the complement (seen domain), then the model tends to learn a function that interpolates correctly the data on the seen domain, and on the unseen domain, the paper characterizes which function the Transformer learns after l2-loss training: the function that has least degree profile in the Fourier spectrum (i.e., least Fourier L2 norm on the high-degree coefficients). This is not necessarily a complex function in the Fourier domain in terms of having for instance large noise sensitivity, as in fact the Transformer minimizes the energy or high degree monomials, but it can produce a complex function in the following sense:
> > >
> > > Consider as an example a target function on many variables but simple in the sense that it is a symmetric decision tree of low depth on few active variables, such as the majority on the first 3 coordinates x1x2+x2x3+x1x3 (or (z1+z2+z3-z1z2z3)/2 in Fourier domain, where zi=(-1)^xi). Assume now that the unseen domain is captured by the condition that we never see x1,x2 to be both 1. Then the transformer will learn a function close this: z1/2+z2/2+z3-z1z3/2-z2z3/2 . The explanation? The paper shows the latter function has least Fourier degree profile and still interpolates the seen data correctly. However this function is more "complex" than the original target function in the sense the Transformer want to both interpolate the seen data and minimize the Fourier degree profile with the unseen "holes", producing weird coefficient mixtures like above that break for instance the symmetry of the function and also loose for instance the Boolean nature of the function output in this case.
> > >
> > > 2. The Boolformer can indeed mitigate this issue; so the introduction gives a valid motivation, but the reviewer is also correct that this first paper about Boolformer does not investigate this specifically and in fact our contributions do not list this; this is listed in our future directions. Instead this paper focuses on more classic noise setting such as limited sample complexity or label noise. This is not the same type of "noise" setting captured by Abbe et al (with the above "generalization on the unseen (GOTU)" setting, that is more about OOD. In this paper we focus on showing that Boolformers do fairly well on specific benchmarks and concrete applications, with a valuable tradeoff in accuracy/time complexity. However the two problems are also not completely unrelated, as the reasons for which Boolformers perform fairly well on these specific tasks and why they would help on GOTU is expected to be the same: the power of Boolformer is that we can better "control" the implicit bias of the model through the data generation process. Let us clarify this:
> > >
> > > In the Boolformer setting, we *first sample an analytical function that has structurel properties that we believe are useful for the task at end*. For instance, we sample low-depth or low-width formula trees, possibly with other features such balanced degrees. By doing so, we can "engineer" the types of analytical formula that we want the model to have biases towards. Then we produce samples from such functions (to create the training set) and then we ask the Boolformer to produce a *token sequence that describes directly the analytical formula*. First one can see that this forces the function learned by the Boolformer to have Boolean output for instance, avoiding already that holes in the sampling domain could break this. Second, we can hope that the "simplicity" here is no longer with respect to the "least Fourier degree profile" (as described above) but w.r. to the generative model bias, which can be completely different (hopefully better but also possibly bad if the generative model is bad).
> > >
> > > So in that sense it is not accurate to say that "there is no redesign from a Fourier spectrum perspective or other robustness innovations"; there is indeed due to the above reason.

---

> ### Author Response · Authors · 2023-11-22
>
> Does this clarify our answers to your questions?

---

> ### Comment · Reviewer_BZRx · 2023-11-23
> **Still not very convincing**
>
> The reviewer thanks the authors for their additional explanations.
>
> On the other hand, the authors simply adopted an existing Transformer-based architecture for symbolic regression to learn boolean functions.
> As for the sampling of the dataset, since it is a synthesized dataset, it seems quite natural to sample what is useful for the target task.
> In conclusion, neither in terms of learning Boolean functions nor in terms of the robustness of the Transformer architecture, it does not bring any technical novelty, but rather it is equivalent to saying that it uses existing knowledge in a field of symbolic regression, which is strongly related to this paper.
>
> Another possibility for the authors regarding technical novelty is the embedder part. However, this is also adopted from (Kamienny et al., 2022), as stated in the paper, and is very similar. Additionally, (Kamienny et al., 2022) proposes an end-to-end symbolic regression method using Transformer. Again, there is no technical novelty beyond a combination of an existing method and a different problem.

---

### Official Review · Reviewer_QDAh · 2023-10-31

**Soundness:** 3 good
**Presentation:** 4 excellent
**Contribution:** 3 good
**Rating:** 8
**Confidence:** 3

**Summary:**

Authors employ Transformer architecture to train a model capable of inferring a logical function based on a truth table.

Authors synthetically produce a dataset to train such a model.

Authors do evaluation in a noiseless and noisy setting.
Noiseless means that the full truth table is available.
Noisy means that a partial truth table is available, and some bits can be flipped with a small probability.
They show that the transformer effectively learned to reconstruct boolean functions.
They also show that it works in a noisy setting.
They evaluate their model in a realistic setting of gene regulatory networks, and showcase the use of their model;
its performance is comparable to other methods while exhibiting much faster inference speed.

**Strengths:**

- Well written and structured
- Clear motivation
- Authors will open source the implementation

**Weaknesses:**

- It would be good to have more examples of real-world usages of the method. Speed (inference) superiority is great, but maybe speed is not even a concern in the domains that the technique is intended to be applied.
- It would be good to have some analysis on which type of tasks is solvable by this method vs. others.
- Does not generalize to larger formulas.

**Questions:**

- Can you explain better Figure 17 from Supplementary Material which displays embeddings?
    - (a) Which part of transformer do you extract for the shown embedding vectors (is it only the last token state, or all tokens, etc.)?
    - (b) What exactly are the inputs that you use to construct shown embedding, and why do you make such choice?

- Figure 1. Denote that what is shown is output of your model. Figure title is misleading.
- Page 3. "in the sections below" -> "in the following sections".
- Page 4. Maybe add that Smax <= Dmax
- Page 5. D refers to dimensionality of logical input. Later in this page, it refers to a set of input-output pairs (if I understand correctly). Use a different letter.
- Fig 7, part a. Readability can be improved.

---

> ### Author Response · Authors · 2023-11-19
>
> We thank the reviewer for their very positive feedback. We will address the presentation aspects.
>
> About the type of tasks solvable by this method : we agree that the paper can benefit from further clarifications. Generally speaking, our method is well-suited for tasks which (i) concern tabular data (structured features) and (ii) where the mapping from input to target can be expressed by a Boolean formula of reasonable complexity. This is the reason we considered the PMLB database, which is arguably the most extensive set of problems with tabular data. As briefly discussed in the results paragraph of Sec 4.2, we see that our method shines particularly in logical tasks, and is less good for other types of datasets. We will extend the discussion on this.
>
> Concerning the question on the embeddings: thanks for noticing that this deserves clarifications. We display the outputs of the “embedder” module, which is a fully-connected network which maps D-dimensional vectors to a single embedding. The inputs are all the points in the 10-dimensional hypercube, sorted by color from 0000000000, 0000000001, 0000000010… 1111111111.

---

> ### Comment · Reviewer_QDAh · 2023-11-21
>
> Would you mind clarifying more Appendix H (Having a more detailed description in it.)?
> I would suggest taking one small image from Figure 16 and showing it separately; including the color diagram mapping color to value (e.g., 0-1).
>
> > For example, for the 4-digit multiplier, some attention heads have hadamard-like structure (e.g. head
> 5 of layer 6), some have block-structured checkboard patterns (e.g. head 12 of layer 4), and many
> heads put most attention weight on the final input, 1111, which is more informative (e.g. head 6 of
> layer 3).
> >> By looking at the image, it seems to me that most of the attention is on the current input; as there's a diagonal pattern.
>
> Figure 17, embeddings, seem not to be grouped by the number value; as within a cluster there's many points with a different color. What do you think clusters represent?
>
> What is the value you use for Demb, and where in the paper do you show this information?

---

> > ### Author Response · Authors · 2023-11-22
> > **Response**
> >
> > Thanks again for your thoughtful suggestions are your careful review!
> >
> > - Would you mind clarifying more Appendix H (Having a more detailed description in it.)? I would suggest taking one small image from Figure 16 and showing it separately; including the color diagram mapping color to value (e.g., 0-1).
> >
> > Indeed, the description of the figures in Appendix H are a bit hasty, and taking out one of the subplots is a very good idea.
> > We will add the following clarifications: "Each attention map is an $N\times N$ matrix, where $N = 2^4 = 16$ is the number of input points. The element $(i,j)$ represents the attention score between tokens $i$ and $j$, and is marked by the colormap, from blue (0) to yellow (1). Here the tokens are ordered from left to right by lexicographic order: : 0000, 0001, 0010, ..., 1111.
> > In this particular order, many interesting structures appear. For example, for the 4-parity function, the anti-diagonal attention map of (head 8, layer 7) indicates that the model compares antipodal points in the hypercube: (0000, 1111), (0011, 1100)..."
> >
> > - "By looking at the image, it seems to me that most of the attention is on the current input; as there's a diagonal pattern."
> >
> > Very sorry about this mistake: we updated the figure and forgot to update the text. We will update with the following: "For example, for the 4-digit multiplier, some attention heads have hadamard-like structure (e.g. heads 3,4,5 of layer 8), some have block-structured checkboard patterns (e.g. head 12 of layer 5), and many heads put most attention weight on the final input, 1111, which is more informative (e.g. head 15 of layer 3)."
> >
> > - "Figure 17, embeddings, seem not to be grouped by the number value; as within a cluster there's many points with a different color. What do you think clusters represent?"
> >
> > Thank you for bringing up this interesting question and reading all the way to the end of the appendix! Given how challenging it is to interpret a 16-dimensional hypercube simply with a t-SNE representation, we preferred not to speculate too much about this figure. But indeed, there are some very interesting trends: the points of the hypercube are hierarchically organized into 4 clusters, each containing 4 subclusters, each containing 4 subclusters... at which point the points are superimposed and indistinguishable. Motivated by the reviewer's question, we looked into this in more detail by investigating what happens in the 5-dimensional hypercube, but were unable to extract a more detailed logic. Here is a figure of what we obtain, in case the reviewer has an idea for an interpretation: https://www.dropbox.com/scl/fi/feb20b9mn1oukhn7ki3ov/embeddings.pdf?rlkey=ly44fylgluytsfz8x8fciy0cw&dl=0.

---

> ### Comment · Reviewer_QDAh · 2023-11-23
>
> Thanks for the clarifications!
>
> Just to clarify. For the 5-dimensional hypercube, you used the same Embedder (embedding model) as for the 10-dimensional hypercube in the appendix?
>
> Does embedder have the maximum dimension it supports, or it's a fixed dimension?
>
> Also, what's the meaning of different colors in the image you sent?

---

> > ### Author Response · Authors · 2023-11-23
> >
> > Yes! The embedder has a maximum dimension $D_{max}$, and pads all inputs to $D_{max}$.
> > In the image I sent, like in the appendix, we colored according to the Hamming weight (number of non-zero bits) of the input, from 0 (blue) to 5 (yellow).
> >
> > By the way, we forgot to answer your question on the embedding dimension: it is 512, as stated in section 2.3.
> > Thanks again for your interest!

---

### Official Review · Reviewer_X8df · 2023-11-01

**Soundness:** 3 good
**Presentation:** 3 good
**Contribution:** 2 fair
**Rating:** 6
**Confidence:** 4

**Summary:**

The paper investigates end-to-end symbolic regression of Boolean functions. The authors show that the overall performance of Boolformer is comparable to classical machine learning in this particular field, with the benefits that Boolformer can be faster and provide interpretable solutions.

**Strengths:**

- The paper is well written and easy to follow.
- Careful description of the generation of synthetic data, and a good analysis of the possible bias included.
- Empirical evaluation establishes the effectiveness of the approach.
- A good section of limitation addressing some of my concerns (e.g., not being able to deal with large formulas) which would otherwise go to the weakness below.

**Weaknesses:**

- While there are some engineering for the embedder, the rest of the approach seems quite standard and straightforward (which is not necessarily a bad thing).
- It might not be that surprising that Boolformer is faster on GRNs tasks. After all, it has been trained for a long time and the training data could have covered what it needed in these tasks. I am curious, however, is there a similar comparison of efficiency in the noiseless regime?

**Questions:**

See above.

---

> ### Author Response · Authors · 2023-11-19
>
> We thank the reviewer for their positive feedback.
> Concerning the comparisons in the noiseless regime, we would like to defer the reviewer to the general comments.

---

### Author Response · Authors · 2023-11-19
**General response on comparisons aspects and general comments about novelty**

We would like to add a general comment to our individual responses, as two of the reviewers discussed comparisons to other approaches in the noiseless setting.

We would like to emphasize that the learning of Boolean/logic functions using deep learning, in particular using Transformers, has been essentially exclusive to the setting in which the loss to the target label is minimized. In the large scale model regime, this can give reasonably good performance but it has two major downsides: (i) it produces black-box predictors which are impossible to explain; (ii) it generalizes well in-distribution but can generalize poorly out-of-distribution [1]. In this work, by adopting the symbolic regression setting and requiring the Transformer to directly output a Boolean function in analytical form, *we mitigate both challenges by outputting explainable functions (see e.g. the examples of decision trees provided in the paper) and by better controlling the “simplicity” bias with the synthetic data generation (this mitigates in particular the Fourier min-degree profile described in [1] for classical Transformers training).* To the best of our knowledge, this is the first work that generates symbolic regression on Boolean functions using neural network/Transformer training, and thus, it is hard to compare it directly to any other analogous methods.

Further, while the results of the noiseless section do not include direct comparisons, as stated in the limitations section, the applicability of our model to predict exact expressions is anyway bottlenecked by (i) the number of inputs it can take and (ii) the number of gates in the predicted expression, which both grow exponentially with the input dimension. Hence, the section on the noiseless setting should be understood more as a “proof of concept” and an investigation into the capabilities and limitations of our model than a practically motivated endeavor. We do not claim that Boolformer solves the Minimal Circuit Size Problem, and will state this more clearly in the camera ready version.

We emphasize that the main contribution of our work, and the most practically relevant one, lies in the noisy regime. For this section, we did our best to choose the most relevant baselines for comparison : for tabular datasets such as PMLB, it is known that tree-based methods such as random forests achieve SOTA performance (see for example [2]), whereas for GRN inference, we selected the most recent benchmark available for boolean inference [3]. Although these baseline methods may be different in nature from ours, we believe they are the most relevant comparison points, given the absence of other approaches for symbolic regression of boolean functions.

Finally, as models get larger and larger (in the classical label loss reduction setting), their explainability gets harder and harder. In the Boolean target case, one could try and take a large model and still estimate the Fourier coefficients with random sampling to find an analytical expression; but there are exponentially many Fourier coefficients to estimate and these could have similar magnitudes (so it is non trivial to sort/truncate). *However, with Boolformers, we can circumvent this undesirable trend:* the number of model’s parameters can be as large as desired for the optimization, and this does not prevent us from requiring the Boolformer output to be as simple of an analytical expression as possible. Whether such an analytical expression will also provide good prediction will depend on the quality of the synthetic formula generator for the tasks at hand; but this is putting the explanation first in a sense. We believe that this is valuable for many tasks, and an approach that was missing in the current paradigm for logic/Boolean function inference (the contribution of our work lies thus also in its timeliness).


[1] Abbe, Emmanuel, et al. "Learning to reason with neural networks: Generalization, unseen data and boolean measures." Advances in Neural Information Processing Systems 35 (2022): 2709-2722.

[2] Grinsztajn, Léo, Edouard Oyallon, and Gaël Varoquaux. "Why do tree-based models still outperform deep learning on typical tabular data?." Advances in Neural Information Processing Systems 35 (2022): 507-520.

[3] Žiga Pušnik, Miha Mraz, Nikolaj Zimic, and Miha Moškon. Review and assessment of boolean approaches for inference of gene regulatory networks. Heliyon, pp. e10222, 2022.

---

### Meta-Review · Area_Chair_MMYL · 2023-12-03

**Metareview:**

The paper introduces Boolformer, a Transformer architecture for end-to-end symbolic regression of Boolean functions. Overall, the reviewers have mixed opinions, ranging from reject to accepts. The main downside pointed out by two reviewers is the limited novelty. The architecture of Boolformer follows standard arguments for symbolic regression, in particular also making use of tons of example formulas that can be generated for SAT. The empirical evidence, however, is interesting, though th  performance of Boolformer is still very similar on average to logistic regression, and even slightly below that of a random forrest. So, in the words of one of the reviewers "the paper is nice but, at this stage, the novelty is too limited for a ICLR publication."

**Justification For Why Not Higher Score:**

As said, the paper is nice but, at this stage, the novelty is too limited for a ICLR publication.

**Justification For Why Not Lower Score:**

N/A

---

### Decision · Program_Chairs · 2024-01-16

Reject